# DNA-PK and the TRF2 iDDR inhibit MRN-initiated resection at leading-end telomeres

Logan R. Myler [1,5], Beatrice Toia [2,5], Cara K. Vaughan[3], Kaori Takai[1], Andreea M. Matei[2], Peng Wu [1,4], Tanya T. Paull [3], Titia de Lange [1] ✉ & Francisca Lottersberger [2] ✉

Telomeres replicated by leading-strand synthesis lack the 3′ overhang required for telomere protection. Surprisingly, resection of these blunt telomeres is initiated by the telomere-specific 5′ exonuclease Apollo rather than the Mre11–Rad50–Nbs1 (MRN) complex, the nuclease that acts at DNA breaks. Without Apollo, leading-end telomeres undergo fusion, which, as demonstrated here, is mediated by alternative end joining. Here, we show that DNA-PK and TRF2 coordinate the repression of MRN at blunt mouse telomeres. DNA-PK represses an MRN-dependent long-range resection, while the endonuclease activity of MRN–CtIP, which could cleave DNA-PK off of blunt telomere ends, is inhibited in vitro and in vivo by the iDDR of TRF2. AlphaFold-Multimer predicts a conserved association of the iDDR with Rad50, potentially interfering with CtIP binding and MRN endonuclease activation. We propose that repression of MRN-mediated resection is a conserved aspect of telomere maintenance and represents an ancient feature of DNA-PK and the iDDR.

TRF2, or TERF2, protects mammalian telomeres by forming a telomere loop (t-loop) structure in which the 3′ single-stranded (ss) overhang invades the duplex part of the telomeres. This architectural change in the telomeric DNA has been proposed to prevent ataxia-telangiectasia mutated kinase (ATM) signaling by denying the double-strand break (DSB) sensor of the ATM pathway, Mre11–Rad50–Nbs1 (MRN), access to the telomere end. In addition, the t-loop structure is proposed to render telomeres impervious to classical non-homologous end joining (c-NHEJ)[1,2] by preventing the loading of the DNA-dependent protein kinase (DNA-PK), comprising the Ku70 and Ku80 heterodimer (Ku70/80) and DNA-PK catalytic subunit (DNa-PKcs), onto telomere ends.

T-loop formation requires the presence of a 3′ overhang. However, after DNA replication, telomeres duplicated by leading-strand DNA synthesis are presumably blunt and require 5′-end resection to regain the 3′ overhang. At DSBs, resection is initiated by MRN and CtIP, whose endonuclease activity nicks the 5′ strand at -15–45 nucleotides (nt) from the break and then uses the 3′ exonuclease activity of MRN to generate a short overhang[1]. This initial resection is required for long-range resection by the Exo1 exonuclease as well as DNA2, which digests 5′ single-strand DNA (ssDNA) generated by the WRN or BLM RecQ helicases. Nonetheless, MRN–CtIP is not used to generate 3′ overhang at telomeres, possibly to avoid MRN-dependent ATM activation, raising the fundamental question of how MRN is kept inactive at newly replicated blunt telomeres. At budding yeast telomeres, the resection function, and possibly the checkpoint function, of the MRN ortholog, MRX, is inhibited by the interaction of the telomeric Rif2 protein with Rad50 (refs. 3–5). Rif2 is found only in some species of budding yeast, but, interestingly, TRF2 carries

[1]Laboratory for Cell Biology and Genetics, The Rockefeller University, New York, NY, USA. [2]Department of Biomedical and Clinical Sciences, Faculty of Medicine and Health Sciences, Linköping University, Linköping, Sweden. [3]Department of Molecular Biosciences, University of Texas at Austin, Austin, TX, USA. [4]Present address: Department of Pediatrics, Stanford University School of Medicine, Stanford, CA, USA. [5]These authors contributed equally: Logan R. Myler, Beatrice Toia. ✉e-mail: delange@mail.rockefeller.edu; francisca.lottersberger@liu.se

an unrelated Rad50-binding module, the inhibitor of the DNA-damage response (iDDR) motif[6]. At dysfunctional telomeres, the iDDR minimizes the accumulation of the DNA-damage factor 53BP1 (ref. 6), but its role at functional telomeres has not been established.

In amniotes, the 5′ exonuclease Apollo (also known as Dclre1b or SNM1B) has evolved a YxLxP motif in its carboxy terminus that allows it to bind to the TRFH domain of TRF2 through the interaction with a region surrounding F120 (refs. 7–11). TRF2-bound Apollo is thought to initiate 5′-end resection at leading-end telomeres to allow subsequent long-range resection by Exo1 (ref. 12). When Apollo is deleted or prevented from binding to TRF2 (for example, when TRF2-F120A is used to complement deletion of TRF2), leading-end telomeres do not regain their normal 3′ overhangs and are vulnerable to end joining[13–17]. Leading-end telomere fusions are abolished by deletion of Ku70, suggesting that they are mediated by c-NHEJ[13,15]. In addition, Apollo deficiency leads to activation of ATM signaling at a subset of telomeres, presumably the leading-end telomeres[16,17]. Because MRN is a requirement for ATM signaling[18–21], MRN must be associated with the unprocessed leading-end telomeres.

Therefore, leading-end telomeres deprived of Apollo are a powerful tool to investigate how MRN can activate ATM signaling without initiating resection at blunt telomere ends. Here we show that the answer lies in the iDDR domain of TRF2, which prevents MRN–CtIP-dependent resection in vivo and inhibits its endonuclease activity in vitro while having no effect on MRN exonuclease activity. AlphaFold-Multimer modeling suggests that the inhibition is due to interaction of iDDR with the ATPase domain of Rad50, a mechanism analogous to the inhibition of MRX by Rif2. AlphaFold modeling also suggests that the iDDR–Rad50 interface overlaps with the binding site of CtIP, possibly explaining the inhibition of the CtIP-dependent endonuclease activity of MRN but not its exonuclease activity. We also show that resection at leading-end telomeres lacking Apollo is inhibited by DNA-PK, such that in the absence of DNA-PK, compensatory resection results in protected telomeres that do not undergo fusion. This result indicates that the previously noted dependence of telomere fusions on Ku70 is not related to the role of DNA-PK in c-NHEJ but is due to its ability to prevent resection when Apollo is absent. In agreement, we find that the telomere fusions in cells lacking Apollo are mediated by alt-EJ and are independent of c-NHEJ factor ligase IV (Lig4).

## Results

### Alt-EJ mediates fusion of telomeres lacking Apollo

In agreement with previous reports that the leading-end telomere fusions do not occur at blunt newly replicated leading-end telomeres in cells lacking Ku70 (refs. 13,15), no telomere fusions were induced upon Cre-mediated deletion of Apollo from cells lacking Ku70, DNA-PKcs or both (Fig. 1a,b and Extended Data Fig. 1a–c). Although these results could be interpreted to mean that the leading-end telomeres are joined by c-NHEJ, this is not the case because the fusions were not dependent on Lig4 (Fig. 1c,d).

We therefore tested the role of alt-EJ in the telomere fusion events. Olaparib inhibition of poly(ADP-ribose) polymerase 1 and 2 (PARP1/2), which mediate the early steps of alt-EJ[22], resulted in a significant reduction in the telomere fusions induced by Apollo deletion (Fig. 1e,f). Similarly, short hairpin RNAs (shRNAs) targeting Ligase III (Lig3)[23] or DNA polymerase theta (PolQ) caused a significant decrease in telomere fusions in cells lacking Apollo (Fig. 1g,h and Extended Data Fig. 1d,e). These data demonstrate that alt-EJ, rather than c-NHEJ, is a major mode of production of blunt telomere fusions and support previous reports indicating that, unlike c-NHEJ, alt-EJ can engage telomeres despite the presence of TRF2 (refs. 24,25).

### DNA-PK loss restores telomere overhang in absence of Apollo

The finding that alt-EJ is responsible for the fusion of blunt telomere ends raised the question of why these fusions are not observed in the absence of DNA-PK. As telomere fusions in Apollo-deficient cells are thought to be a consequence of a resection defect, we explored the possibility that loss of DNA-PK unleashes compensatory resection that restores the 3′ overhang. A role for DNA-PK as a repressor of resection would be in line with Ku70/80 blocking resection in yeast[26–29]. Indeed, after Apollo deletion, cells lacking Ku70 or DNA-PKcs did not show the decrease in the overhang signal observed in DNA-PK-proficient cells (Fig. 2a,b). By contrast, depletion of Lig3 or PolQ did not have this effect, and cells lacking alt-EJ showed the expected reduction in the 3′ overhang signal upon Apollo deletion (Fig. 2c,d), although leading-end telomere fusions were reduced. Similarly, and consistent with the telomere fusion phenotype, the lack of Lig4 did not affect the reduction in the overhang due to Apollo deletion (Extended Data Fig. 2a,b). Conditional deletion of *Ku70* from otherwise wild-type mouse embryonic fibroblasts (MEFs) did not alter the 3′ overhang signal (Fig. 2e,f and Extended Data Fig. 2c,d), indicating that the effect of Ku70 on telomere resection is apparent only when Apollo is absent. Thus, the presence of DNA-PK appears to inhibit Apollo-independent resection at newly replicated leading-end telomeres. Without this inhibition by DNA-PK, 3′ overhang formation and telomere protection are generated independently of Apollo.

### Nbs1 represses telomere fusions in the absence of DNA-PK and Apollo

To test how DNA-PK represses resection at telomeres, we targeted Nbs1 with CRISPR–Cas9 (Fig. 3a). Bulk targeting of Nbs1 did not induce telomere fusions in DNA-PK-null MEFs expressing Apollo (Fig. 3b,c). However, when DNA-PKcs and Apollo were both absent, Nbs1 targeting increased the frequency of leading-end telomere fusions (Fig. 3b,c), suggesting that DNA-PK represses resection by MRN. The frequency of telomere fusions induced by CRISPR–Cas9 targeting of Nbs1 in DNA-PKcs-null cells was lower than that observed in DNA-PK-proficient Apollo-knockout cells (Fig. 3b,c). It is possible that the CRISPR–Cas9 targeting was not sufficient to abort MRN-dependent resection in all cells; however, it is also possible that other nucleases promote resection of the blunt telomeres in the absence of DNA-PK, together or independently of MRN. Indeed, depletion of Exo1, which resects telomeres after replication in normal conditions[12], did not induce telomere fusions (Extended Data Fig. 2e), but caused a significant reduction of the overhang signal when combined with Apollo deletion in DNA-PKcs-deficient cells (Extended Data Fig. 2f,g), suggesting that Exo1 contributes to the resection after it is initiated by MRN.

### The iDDR of TRF2 inhibits MRN–CtIP at leading-end telomeres

The observation of DNA-PK preventing resection at blunt telomeres is surprising, given that DNA-PK promotes MRN–CtIP endonuclease activity at DSBs[30], and it suggests that MRN–CtIP has different access to telomeres than to DSBs. As the iDDR of TRF2 interacts with MRN[6], we asked whether the iDDR affects MRN-initiated resection at telomeres in cells lacking Apollo. We generated conditional *Trf2*[F/F] cells expressing wild-type *Trf2* (WT), the *Trf2*[ΔiDDR] allele (ΔiDDR), the *Trf2*[F120A] allele (F120A), whose product does not bind Apollo[7], or a version of *Trf2* containing both mutations (F120A ΔiDDR) (Fig. 4a and Extended Data Fig. 3a,b). Importantly, removal of the iDDR completely mitigated the effect of Apollo loss, restoring the 3′ overhang signal and abolishing leading-end telomere fusions (Fig. 4b–e), although Chk2 phosphorylation increased (Fig. 4a and Extended Data Fig. 3c). In addition, the deletion of the iDDR by itself caused a small reduction in proliferation and an increase of γ-H2AX foci at telomeres, as well as a trend toward greater telomeric overhang signals for reasons that remain to be determined (Fig. 4b,c and Extended Data Fig. 3d–f).

To establish whether the iDDR acts by controlling MRN, we expressed the same *Trf2* alleles in cells lacking Nbs1 and examined the telomeric overhang signals and telomere fusions after TRF2 deletion. In accordance with the results presented above, in the Nbs1-proficient

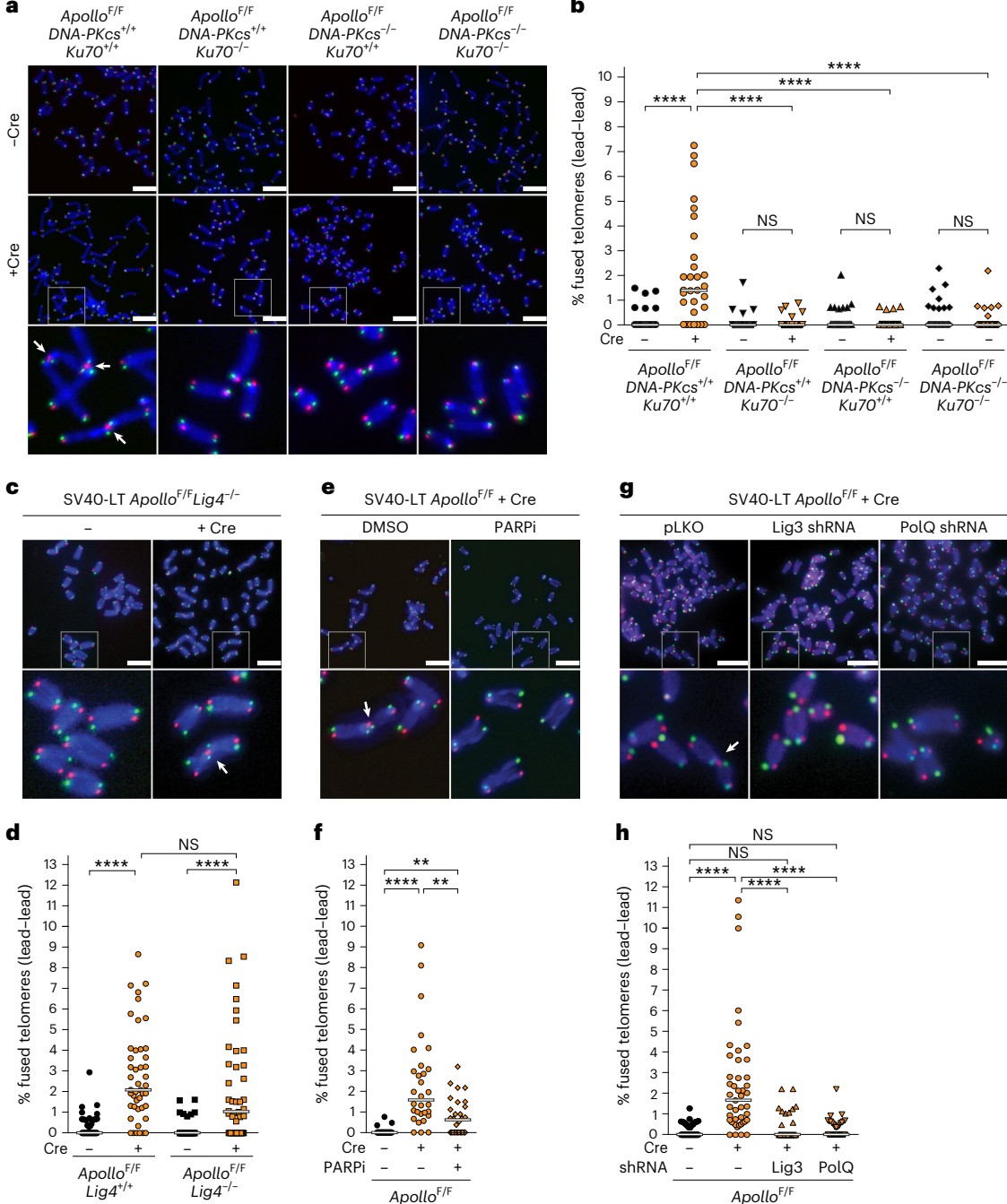

**Fig. 1 | Alt-EJ promotes leading-end telomere fusions due to Apollo deletion.**
**a**, Representative chromosome orientation fluorescence in situ hybridization
(CO-FISH) of metaphase spreads in *Apollo*^F/F^*DNA-PKcs*^+/+^*Ku70*^+/+^, *Apollo*^F/F^
*DNA-PKcs*^+/+^*Ku70*^−/−^, *Apollo*^F/F^*DNA-PKcs*^−/−^*Ku70*^+/+^, or *Apollo*^F/F^*DNA-PKcs*^−/−^*Ku70*^−/−^
MEFs immortalized by simian virus 40 large T antigen (SV40LT) without any
treatment or 96 h after Hit & Run Cre-mediated deletion of Apollo. Leading- and
lagging-end telomeres were detected with Cy3-(TTAGGG)₃ (red) and Alexa-
Fluor-488-(CCCTAA)₃ (green) probes, respectively. DNA was stained with DAPI
(blue). Arrows indicate leading-end telomere fusions. The boxed regions are
enlarged in the bottom row. **b**, Quantification of leading-end telomere fusions
as shown in **a**. Each dot represents the percentage of telomeres fused in one
metaphase. Bars represent the median of fused telomeres in *n* = 30 metaphases
over 3 independent experiments (10 metaphases per experiment). Only fusions
involving two leading-end telomeres (lead–lead) are shown. **c,d**, Representative
micrographs of metaphase spreads in *Apollo*^F/F^*Lig4*^+/+^ and *Apollo*^F/F^*Lig4*^−/−^ MEFs (**c**)

and quantification of leading-end telomere fusions (**d**) before and 96 h after
Hit & Run Cre. Quantification as in **b** for *n* = 45 metaphases over 3 independent
experiments (15 metaphases per experiment). **e,f**, Representative micrographs
of metaphase spreads of *Apollo*^F/F^ MEFs (**e**) and quantification of leading-end
telomere fusions (**f**) 96 h after Hit & Run Cre and/or after 24 h of treatment with
2 μM of the PARP inhibitor olaparib (PARPi). Quantification as in **b** for *n* = 30
metaphases over 3 independent experiments (10 metaphases per experiment).
**g,h**, Representative micrographs of metaphase spreads of *Apollo*^F/F^ MEFs (**g**) and
quantification of leading-end telomere fusions (**h**) 108 h after Hit & Run Cre. Cells
were transduced with empty vector or an shRNA targeting *Lig3* or *PolQ*. for *n* = 45
metaphases over 3 independent experiments (15 metaphases per experiment).
Statistical analysis by Kruskal–Wallis one-way analysis of variance (ANOVA) for
multiple comparisons. Scale bars (**a,c,e,g**), 10 μm. *****P* < 0.0001, ****P* < 0.001,
***P* < 0.01, **P* < 0.05. NS, not significant. See also Extended Data Fig. 1.

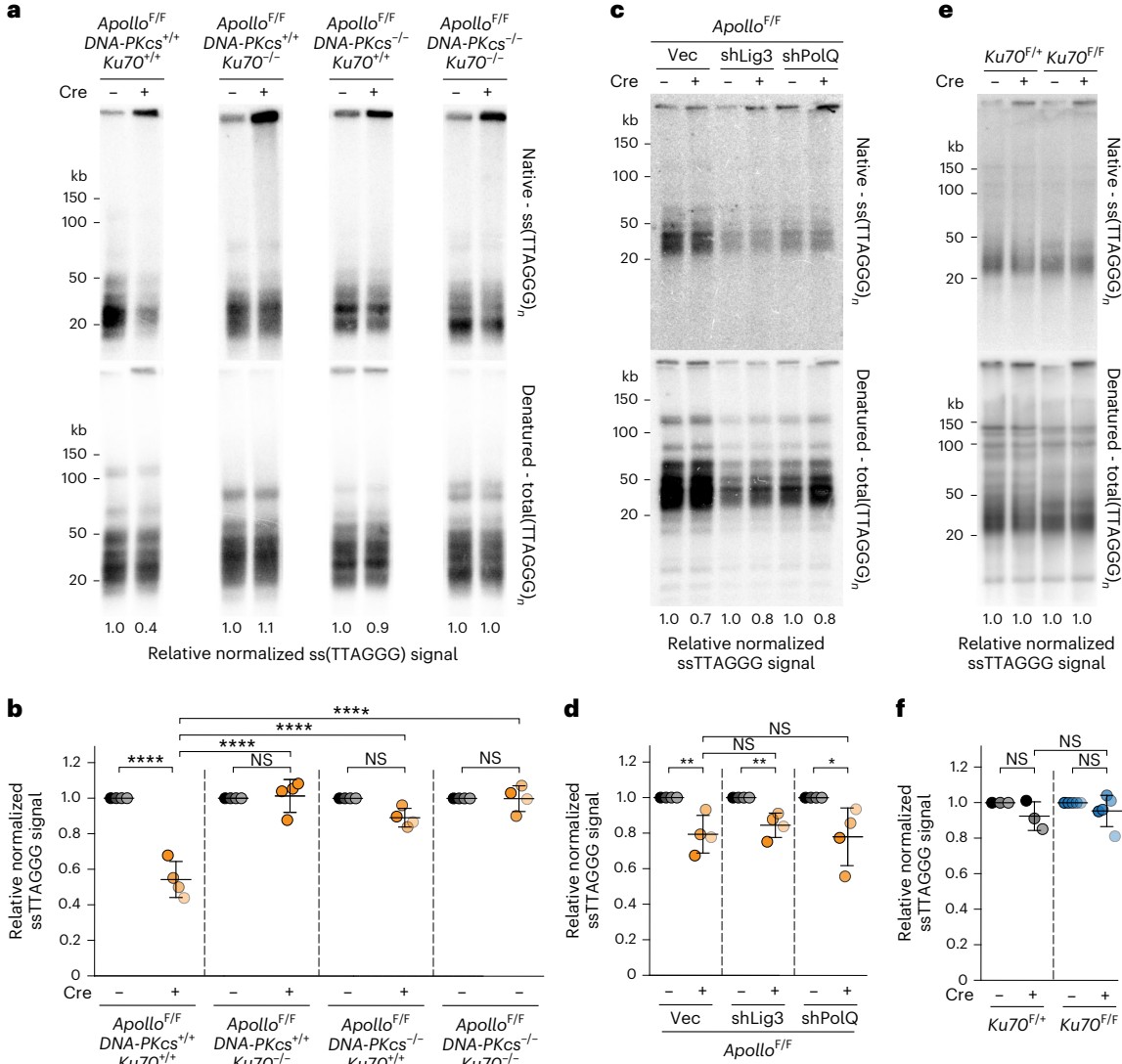

**Fig. 2 | DNA-PK prevents Apollo-independent processing of 3′ telomere overhang. a**, Telomeric overhang assay on SV40LT-immortalized *Apollo*[F/F] *DNA-PKcs*[+/+]*Ku70*[+/+], *Apollo*[F/F]*DNA-PKcs*[+/+]*Ku70*[−/−], *Apollo*[F/F]*DNA-PKcs*[−/−]*Ku70*[+/+] or *Apollo*[F/F]*DNA-PKcs*[−/−]*Ku70*[−/−] MEFs 96 h after Hit & Run Cre-mediated deletion of endogenous Apollo. Top, single-stranded telomeric DNA signal (Native - ss(TTAGGG)₃). Bottom, total telomeric signal (Denatured - total(TTAGGG)₃). The ssTTAGGG signal was normalized to the total telomeric DNA in the same lane. The normalized −Cre value for each cell line is set to 1, and the +Cre value is given relative to 1. **b**, Quantification of the relative overhang signal as detected in **a** for *n* = 4 independent experiments (indicated by different shades), with mean ± s.d. indicated. **c,d**, Telomeric overhang assay (**c**) and quantification (**d**) on SV40LT-immortalized *Apollo*[F/F] MEFs transduced with empty vector or shRNAs targeting Lig3 or PolQ and 108 h after Cre-mediated deletion of endogenous Apollo. *n* = 4 independent experiments. Data are presented as mean ± s.d. **e,f**, Telomeric overhang assay (**e**) and quantification (**f**) of *Ku70*[F/+] (*n* = 3) and two independent *Ku70*[F/F] (*n* = 5) MEFs 96 h after Hit & Run Cre-mediated deletion of endogenous Ku70. *n* = 3 independent experiments. Data are presented as mean ± s.d. Statistical analysis by two-way ANOVA (**b,f**) and two-tailed unpaired *t*-test (**d**). ****$P < 0.0001$, ***$P < 0.001$, **$P < 0.01$, *$P < 0.05$. See also Extended Data Figs. 1 and 2.

cells, absence of the iDDR mitigated the 3′ overhang defect observed in absence of Apollo. By contrast, the deletion of the iDDR from TR2-F120A cells did not improve the processing of leading-end telomeres when Nbs1 was absent (Fig. 5a,b). Similarly, the leading-end fusions at telomeres lacking Apollo were diminished when the iDDR was removed from TRF2 in Nbs1-proficient cells. By contrast, in Nbs1-deficient cells, removal of the iDDR from TRF2 did not affect the leading-end telomere fusions owing to lack of Apollo recruitment (Fig. 5c,d). These results indicate that the iDDR of TRF2 acts through MRN.

**No role for 53BP1**

Because 53BP1 blocks the formation of excessively long 3′ overhangs at dysfunctional telomeres[23,31–33], we tested whether 53BP1 also affected the formation of 3′ overhangs at telomeres lacking Apollo. However, in

53BP1-deficient cells with TRF2 deletion that expressed the F120A allele, there was a reduction of the 3′ overhang signal; for 53BP1-proficient cells, no reduction was observed in cells with the F120A ΔiDDR allele (Extended Data Fig. 4a). Furthermore, 53BP1 status had no effect on the reduction in the 3′ overhang after Apollo deletion (Extended Data Fig. 4b,c), indicating that the iDDR inhibits MRN–CtIP independently of 53BP1.

**In vitro inhibition of MRN–CtIP endonuclease by the iDDR**

Given that the iDDR inhibits MRN in a 53BP1-independent manner, we asked whether it directly affects MRN activity. MRN is an endonuclease that is activated by phosphorylated CtIP to nick the 5′ strand at protein-blocked DNA ends[34–36]. The endonuclease activity of MRN–CtIP can be measured under physiological conditions on a DNA substrate

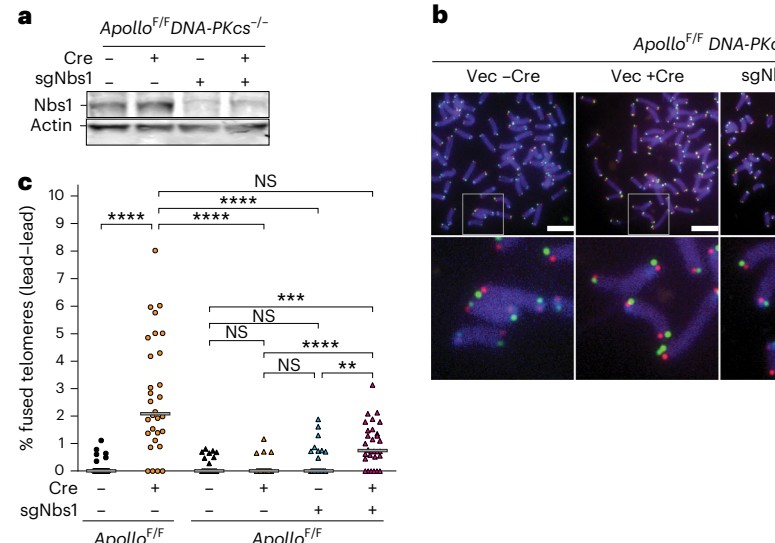

**Fig. 3 | Nbs1 protects from leading-end telomeres fusions in the absence of Apollo and DNA-PKcs. a**, Immunoblots for Nbs1 in SV40LT-immortalized *Apollo*^F/F*DNA-PKcs*⁻/⁻ MEFs, after transduction with Cas9 expression vector with short guide RNA (sgRNA) targeting *Nbs1* or without this sgRNA (Vec) and/or Hit & Run Cre. Actin is shown as a loading control. **b,c**, Representative micrographs of metaphase spreads in *Apollo*^F/F and *Apollo*^F/F*DNA-PKcs*⁻/⁻ MEFs (scale bars, 10 μm) (**b**) and quantification of leading-end telomere fusions (**c**) 96 h after

Cre-mediated deletion of Apollo and/or 48 h after deletion of Nbs1 by CRISPR–Cas9 and the specific sgRNA, as shown in **a**, for *n* = 30 metaphases collected over 3 independent experiments (10 metaphases per experiment). In micrographs, the boxed area is enlarged in the bottom row. The white arrow indicates a leading-end telomere fusion. In the graph, bars represent the median. Statistical analysis by Kruskal–Wallis one-way ANOVA for multiple comparisons. ****$P$ < 0.0001, ***$P$ < 0.001, **$P$ < 0.01, *$P$ < 0.05.

that is bound to DNA-PK. In this setting, MRN generates an endonucleolytic product of approximately 45 nt[30]. Indeed, purified TRF2 inhibited the formation of this product in a dose-dependent manner (Fig. 6a–c). By contrast, the ΔiDDR motif TRF2 mutant did not have this effect, indicating that the iDDR is required for the inhibition of MRN endonuclease activity. MRN also exhibits a 3′–5′ exonuclease activity that is independent of CtIP[37,38]. In contrast with its role in inhibiting the endonuclease activity of MRN, the addition of WT or ΔiDDR TRF2 did not inhibit the exonuclease activity of MRN (Fig. 6d–f). These results show that the iDDR has a specific effect on the endonuclease activity of MRN–CtIP but does not interfere with the DNA-end-binding or exonuclease activity of MRN. The effect of the iDDR is similar to that of Rif2 in budding yeast, which inhibits MRX endo- but not exonuclease activity[3,4].

**AlphaFold predicts the interaction between iDDR and RAD50**
It has previously been shown that the iDDR of TRF2 pulls down Rad50 in co-immunoprecipitation experiments[6]. However, these experiments did not determine which subunit of MRN interacts with TRF2. We investigated the potential interactions of the human iDDR with MRN subunits using AlphaFold-Multimer[39]. AlphaFold-Multimer predicted an interaction between the iDDR and the globular ATPase domain of RAD50 (Fig. 7a–c). The iDDR was predicted with high confidence (pLDDT value) in the top five ranked models (Fig. 7b). No interactions were predicted with MRE11, NBS1 or CtIP, and the iDDR was predicted with low confidence (predicted local distance difference test (pLDDT) value) in these models, suggesting that RAD50 binding orders the domain (Extended Data Fig. 5a). The predicted interaction of the iDDR with RAD50 is highly conserved in metazoans, including both vertebrate TRF2 and invertebrate TRF proteins (Extended Data Fig. 5b,c).

   The telomere-binding proteins Rif2 and Taz1 of budding and fission yeast, respectively (Extended Data Fig. 6a), have recently been proposed to bind to Rad50 via their MRN(X)-inhibitory (MIN) domains (also called the BAT domain in Rif2)[3,5]. AlphaFold-Multimer predicted that Rif2 and Taz1 bind their cognate Rad50 proteins using the same

interface in the ATPase domain used by the iDDR (Extended Data Fig. 6b,c). Despite this analogous binding site, the MIN domains of Rif2 and Taz1 do not show sequence homology and are predicted to be structurally different from the iDDR domain. In fact, the iDDR consists of a cluster of basic residues followed by acidic residues, which create a characteristic drop in local isoelectric point (pI) (Extended Data Fig. 6d,e; ref. 10). On the contrary, the MIN domains of Rif2 and Taz1 have a switched order of basic and acidic residues. This inversion is reflected in their predicted orientation on the surface of Rad50, with the N-terminal residues of the MIN domain pointing toward the coiled-coils, whereas in the iDDR, the C terminus is pointing towards the coiled-coils (Extended Data Fig. 6d). This inverted orientation suggests that these motifs may have evolved independently and that their similar function is due to convergent evolution (Extended Data Fig. 6f), and therefore that an MRN-inhibitory module is a highly selected and ancient feature of telomeres.

**The iDDR is predicted to compete with CtIP for RAD50 binding**
Data from budding yeast suggest that Rif2 inhibits the MRX endonuclease activity by outcompeting Sae2 at its Rad50-binding site[4,34]. Therefore, we used AlphaFold-Multimer to query whether the iDDR might similarly affect the interactions between CtIP and the MRN complex. The 'Sae2-like' C terminus of CtIP was predicted with high confidence to interact with the same surface of RAD50 that the iDDR of TRF2 also interacts with (Fig. 7d–f). Interestingly, this is the region where the RAD50S separation-of-function alterations cluster, which are deficient in CtIP-dependent endonuclease, but not in CtIP-independent exonuclease, activity[40–42]. This region of CtIP is highly conserved across metazoans, consistent with its potential role in regulating the MRN complex (Extended Data Fig. 7a), which is also highly conserved. Additionally, Sae2 and Ctp1 (the fission yeast homolog of CtIP) were predicted to bind to their respective Rad50 proteins in a similar manner (Extended Data Fig. 7b–d). Collectively, the data suggest that the iDDR of TRF2 can prevent CtIP from binding to RAD50, blocking the activation of MRN endonuclease activity (Fig. 7g).

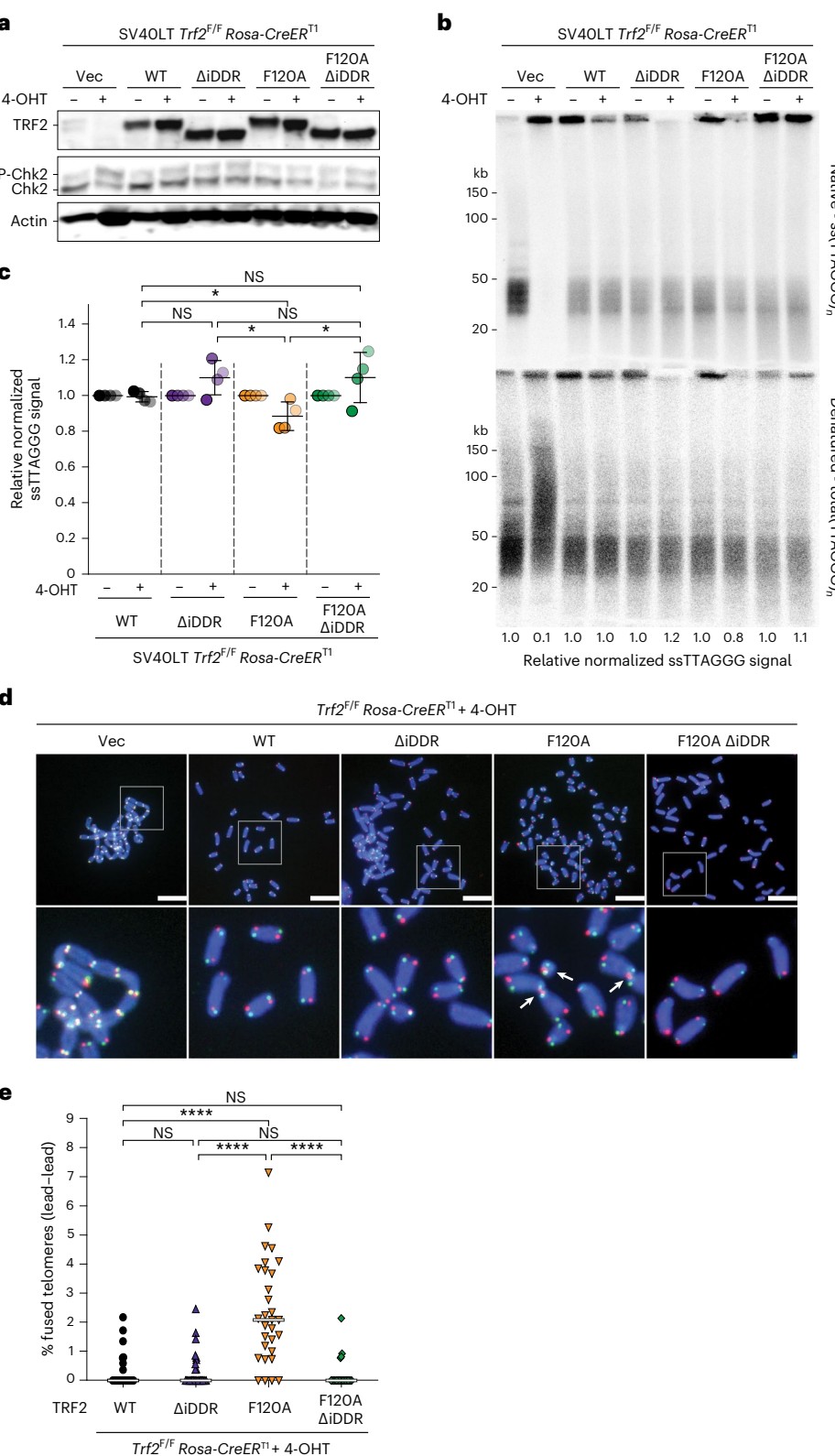

**Fig. 4 | TRF2 prevents Apollo-independent processing of 3′ telomere overhang through the iDDR region. a**, Immunoblot of endogenous and exogenous TRF2 and Chk2 phosphorylation in SV40LT-immortalized *Trf2*[F/][F]*RsCre-ER*[T1] MEFs expressing empty vector (EV) or the *MYC-Trf2* (WT), *MYC-Trf2*[ΔiDDR] (ΔiDDR), *MYC-Trf2*[F120A] (F120A) or *MYC-Trf2*[F120A ΔiDDR] (F120A ΔiDDR) alleles at 96 h after 4-OH tamoxifen (4-OHT)-mediated deletion of endogenous TRF2. Actin is shown as a loading control. **b,c**, Telomeric overhang assay (**b**) and quantification (**c**) from *n* = 4 independent experiments of cells treated as described in **a**. For each *MYC-Trf2* allele, the normalized −Cre value was set to 1,

and the +Cre value was given relative to 1, with mean ± s.d. indicated. Statistical analysis by two-tailed unpaired *t*-test. **d,e**, CO-FISH metaphase analysis (**d**) and quantification of leading-end telomere fusions (**e**) in *Trf2*[F/F]*RsCre-ER*[T1] MEFs expressing empty vector (EV) or one of the *MYC-Trf2* alleles described in **a** 96 h after treatment with 4-OHT. Scale bars, 10 μm. The bars on the graph show the median. Statistical analysis by Kruskal–Wallis one-way ANOVA for multiple comparisons for *n* = 30 metaphases over 3 independent experiments (10 metaphases per experiment). ****$P < 0.0001$, ***$P < 0.001$, **$P < 0.01$, *$P < 0.05$. See also Extended Data Fig. 3.

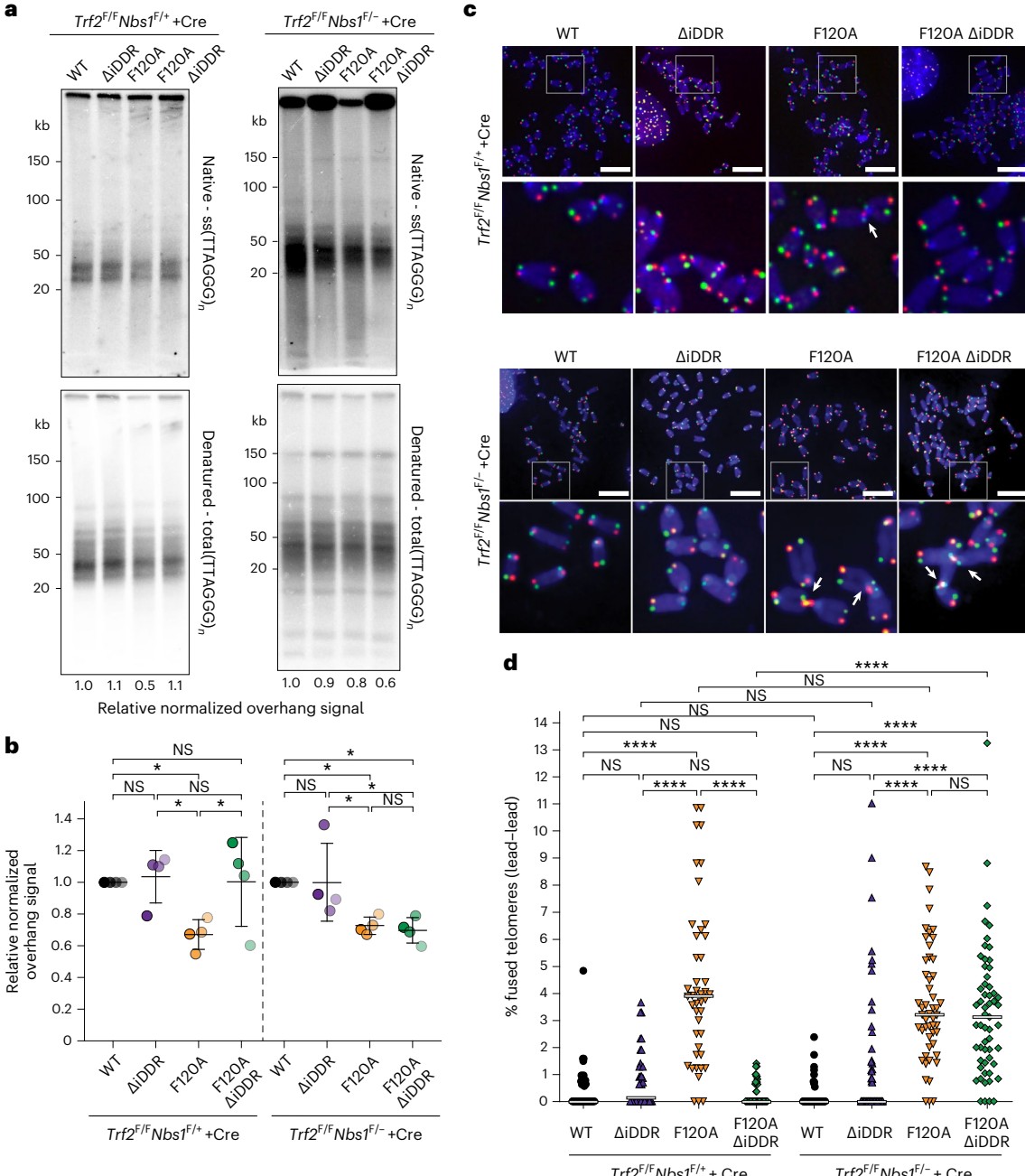

**Fig. 5 | The TRF2 iDDR acts through MRN. a,b,** Telomeric overhang assay and quantification from $n = 4$ independent experiments of $Trf2^{F/F}Nbs1^{F/+}$ and $Trf2^{F/}$ $^{F}Nbs1^{F/-}$ MEFs expressing the indicated $MYC\text{-}Trf2$ alleles 120 h after Cre-mediated deletion of TRF2 or TRF2 and Nbs1. Each cell line was normalized to the WT allele. Data are presented as mean ± s.d. Statistical analysis by two-tailed unpaired $t$-test. **c,d,** CO-FISH metaphase analysis (**c**) and quantification of leading-end telomere fusions (**d**) in $Trf2^{F/F}Nbs1^{F/+}$ and $Trf2^{F/F}Nbs1^{F/-}$ MEFs expressing the indicated

$MYC\text{-}Trf2$ alleles 120 h after Cre-mediated deletion of TRF2 or TRF2 and Nbs1. Scale bars, 10 μm. Graph represents $n = 45, 38, 44, 64, 58, 58, 51, 57$ metaphases over 3 independent experiments (at least 10 metaphases per experiment), with medians. Statistical analysis by Kruskal–Wallis one-way ANOVA for multiple comparisons. ****$P < 0.0001$, ***$P < 0.001$, **$P < 0.01$, *$P < 0.05$. See also Extended Data Fig. 4.

## Discussion

These results reveal the ability of the TRF2 iDDR and DNA-PK to independently inhibit all MRN-mediated resection at blunt-ended telomeres, although they resemble DSBs (Fig. 7h). When telomeres are replicated in the absence of Apollo, the leading-strand DNA-synthesis products lack the protective 3′ overhang and become vulnerable to alt-EJ. These fusion events depend on the ability of DNA-PK to prevent long-range resection. However, it has been puzzling why MRN–CtIP does not promote DNA end processing by removing DNA-PK from these telomeres, as it does at DSBs. Here we show that the iDDR of TRF2 blocks MRN–CtIP

endonuclease activity in vitro and prevents MRN from acting at blunt leading-end telomeres in vivo. Our structure-prediction data suggest that the iDDR acts by preventing the formation of the MRN–CtIP complex. Because MRN–CtIP is not active at telomeres bearing the iDDR, DNA-PK can persist at these ends, leading to complete absence of resection. When the iDDR is removed, MRN–CtIP is active and can cleave DNA-PK off the telomere ends. Interestingly, when DNA-PK is absent, MRN can still engage the ends, leading to long-range resection, although the endonucleolytic activity of MRN–CtIP should be held in check by the iDDR. In both scenarios, resection is activated.

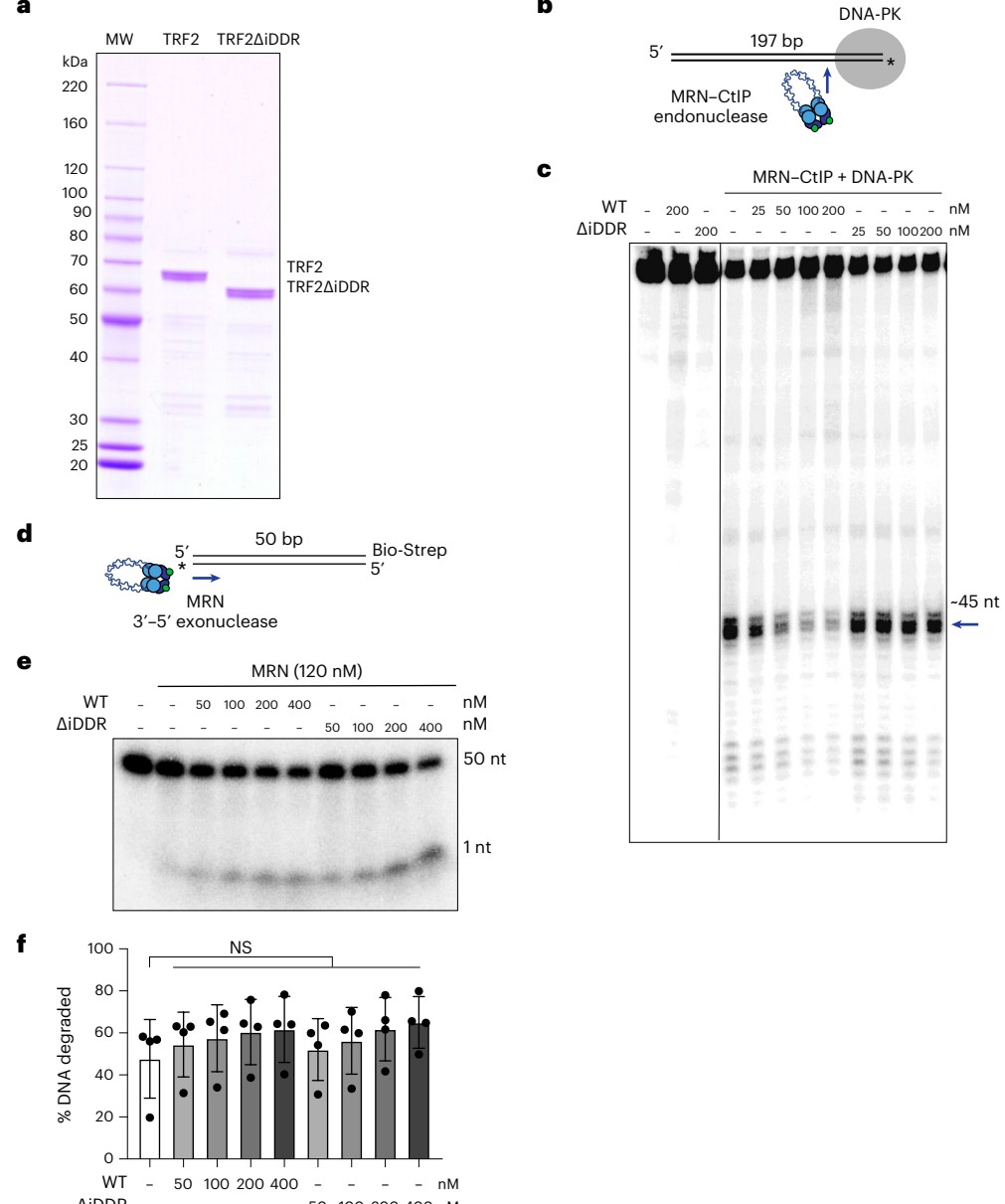

**Fig. 6 | The iDDR of TRF2 inhibits MRN endonuclease activity in vitro.**
**a**, Coomassie-stained SDS–PAGE gel of purified TRF2 proteins. The gel is a representative of three independent protein preparations. WT and ΔiDDR were prepared identically. Protein molecular weight (MW) is shown as control. **b**, Schematic of the MRN–CtIP endonuclease assay with DNA-PK. **c**, MRN endonuclease assay in the presence of WT or ΔiDDR TRF2. MRN (50 nM) was incubated with phosphorylated CtIP (80 nM), DNA-PKcs (10 nM), Ku70/80 (10 nM) and varying concentrations of TRF2 (25, 50, 100 or 200 nM) in the presence of a 5′ $^{32}$P-labeled DNA substrate. The gel is a representative example of two independent replicates. The blue arrow indicates the primary endonucleolytic cleavage product (~45 nt away from the end). **d**, Schematic of the MRN exonuclease assay. **e,f**, MRN endonuclease assay gel (**e**) and quantification (**f**) of $n = 4$ independent replicates in the presence of WT or ΔiDDR TRF2. Data are presented as mean ± s.d. Statistics by two-tailed unpaired $t$-test assuming a Gaussian distribution. ****$P < 0.0001$, ***$P < 0.001$, **$P < 0.01$, *$P < 0.05$.

Paradoxically, this leads to the formation of a protective 3′ overhang independently of Apollo. Importantly, we show that the inhibition of MRN endonuclease activity at eukaryotic telomeres is conserved, owing in part to convergent evolution.

The iDDR is an ancient feature of the TRF subunit of shelterin. It was already present when metazoans emerged, prior to the gene duplication that created TRF1 and TRF2 and long before Apollo–TRF2 binding evolved[10]. By contrast, the genes involved in the ability of mammalian iDDR to minimize the accumulation of 53BP1 at dysfunctional telomeres, including *BRCC3*, *RNF8*, *RNF168* and *53BP1* itself, are not conserved in all metazoans[43]. This argues that the iDDR evolved the ability to affect

53BP1 later as a secondary feature. The argument that inhibiting MRN is the original function of the iDDR is strengthened by the finding that the MRN/X inhibitory modules (called MIN or BAT) are found at telomeres in fungi. Remarkably, these fungal MIN modules interact with the same part of Rad50 that the iDDR interacts with. Because the MIN/BAT have no sequence similarity to the iDDR, are found in proteins that are not orthologous to TRF2 (for example, Rif2) and bind in an inverse orientation compared with the iDDR, we infer that the inhibition of MRN at telomeres represents an example of convergent evolution. Such convergent evolution speaks to the strong selective pressure to preserve the inhibition of MRN at telomeres, despite major changes in the

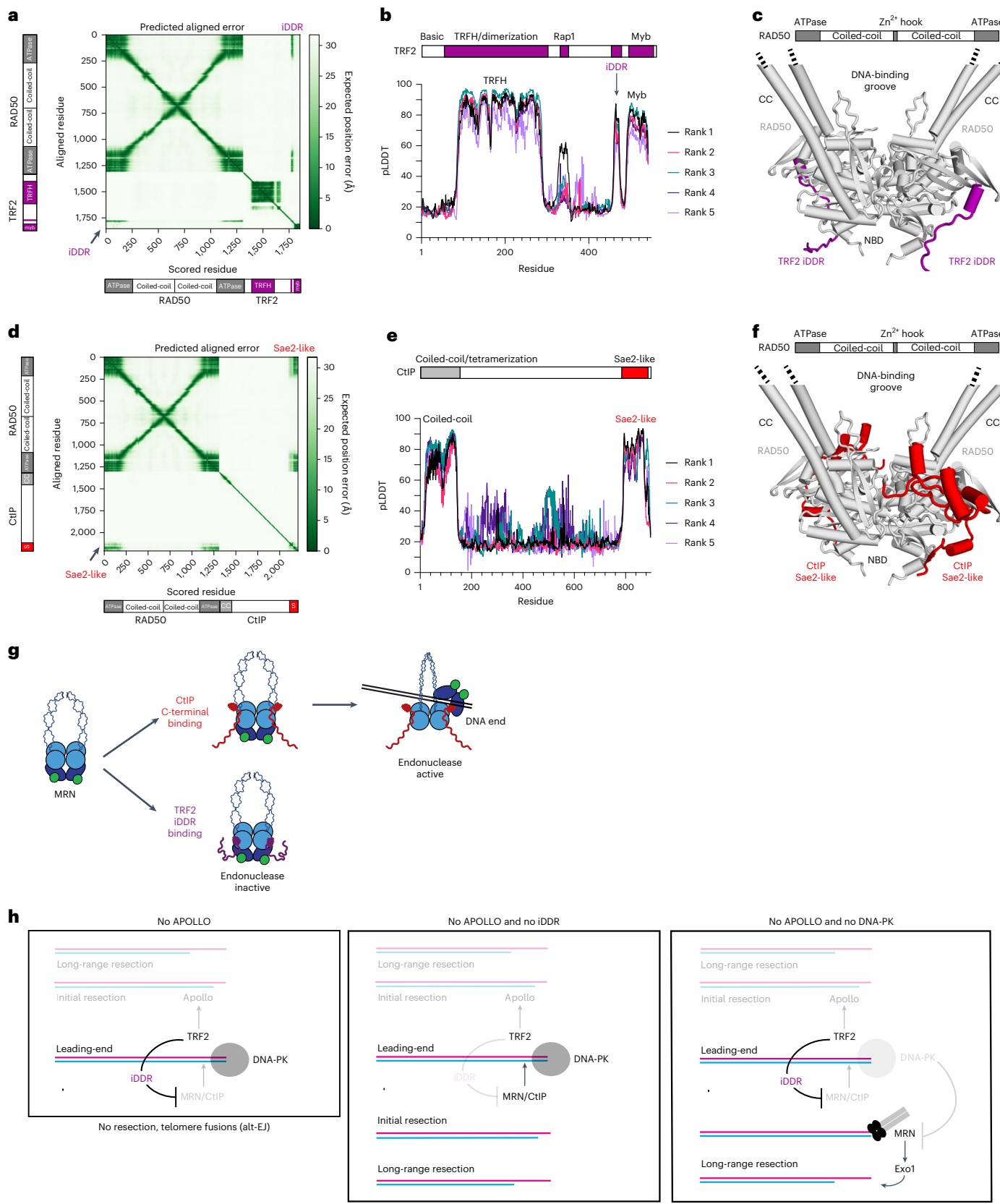

telomere-associated proteins. Why the endonuclease activity of MRN needs to be inhibited at telomeres and/or whether iDDR contributes to direct inhibition of ATM activation at telomeres, as shown in yeast for the MIN/BAT[5] and suggested by high Chk2 phosphorylation in the presence of Trf2$^{F120A\,\Delta iDDR}$, remains to be determined.

DNA-PK blocks the access of the long-range nucleases Exo1 and DNA2/BLM to DNA ends in vitro and suppress DNA end resection in vivo[30,44,45]. However, counterintuitively, the presence of DNA-PK also stimulates MRN–CtIP endonuclease activity in vitro[30]. These results are consistent with the essential role of MRN–CtIP in initiating resection

**Fig. 7 | The iDDR of TRF2 is predicted to compete with CtIP for binding to RAD50. a**, Predicted aligned error (PAE) plot from AlphaFold-Multimer modeling of human RAD50–TRF2. Representative of five ranked models generated with default parameters. **b**, pLDDT plot showing per-residue confidence score across human TRF2 from the ranked RAD50–TRF2 models. **c**, Predicted structure of a human RAD50 dimer with a dimer of human TRF2. The RAD50 coiled-coils were truncated for the dimer model. Only the iDDR of TRF2 is shown. **d**, PAE plot from the AlphaFold-Multimer modeling of human RAD50–CtIP. Representative of five ranked models generated with default parameters. **e**, pLDDT plot showing per-residue confidence score across human CtIP from the ranked RAD50–CtIP models. **f**, Predicted structure of a human RAD50 dimer with the Sae2-like domain of human CtIP. The RAD50–CtIP monomer model was superimposed on the predicted structure of an RAD50 dimer from **c**. **g**, Model for inhibition of MRN

endonuclease activity by the TRF2 iDDR. TRF2 competes with CtIP for binding to RAD50, which stimulates the endonuclease-active state of MRN. **h**, Model for leading-end telomere processing and protection mediated by TRF2 and DNA-PK. TRF2 promotes the 5′-resection of the leading-end telomere by recruiting Apollo. In the absence of Apollo, either owing to Apollo deletion or the lack of recruitment due to TRF2-F120A, the newly replicated leading-end telomere ends cannot be resected and undergo fusion mediated by alt-NHEJ. In the absence of Apollo and the iDDR domain of TRF2, MRN–CtIP initiates resection at DNA-PK-bound leading-end telomeres, leading to telomere protection and the absence of fusions. When both DNA-PK and Apollo are absent, MRN can promote the resection of the free DNA ends, even in the presence of the TRF2 iDDR domain. See also Extended Data Figs. 5–7.

at ends blocked by DNA-PK, such as single-ended DSBs resulting from replication fork collapse[46]. Our data demonstrate that the iDDR inhibits MRN–CtIP from acting as an endonuclease, effectively blocking all MRN–CtIP-mediated resection at the DNA-PK-bound telomeres. Importantly, DNA-PK, which localizes at telomeres in normal conditions[47,48], allows Apollo processing. However, in the absence of DNA-PK, telomeres can be processed independently of Apollo. This resection seems to at least partially depend on Nbs1. Therefore, we propose that MRN can promote the resection of blunt-ended telomeres when DNA-PK is absent, possibly owing to its ability to load other resection factors at the ends, as MRN itself has no 5′ exonuclease activity. We further propose that this loading activity of MRN does not require CtIP, explaining why it is not repressed by the iDDR. In agreement, MRN promotes loading of Exo1 and other long-range resection factors like BLM/DNA2 at DNA ends in vitro and enhances the processivity of Exo1 resection in the presence of RPA[49–51]. Consistent with this, Exo1 is involved in overhang generation in the absence of DNA-PKcs and Apollo, although probably only after the first step of resection is initiated. Whether other resection regulators in addition to MRN and Exo1 gain access to the telomere ends that lack DNA-PK remains to be determined.

The data show that the leading-end telomeres in cells lacking Apollo become joined by PolQ-dependent alt-EJ (also called theta-mediated end joining, TMEJ)[52,53]. This result was unexpected because leading-end telomeres lacking Apollo are presumed to be blunt, whereas alt-EJ joins DSBs with 3′ overhangs. In alt-EJ, PolQ uses microhomologies to anneal 3′ overhangs and then executes templated DNA synthesis to create the substrate for Lig3-mediated ligation. Because PolQ is auto-inhibited at DNA ends with short or no 3′ overhangs[54], it remains to be determined how it acts on the blunt leading-end telomeres. Conversely, it is prudent to ask why alt-EJ does not act on telomeres that have a 3′ overhang that can anneal to form 2 base pairs every 6 nt. Most likely, this is owing to the presence of the POT1 (Pot1a and Pot1b in mice) on the ssDNA, because alt-EJ has been shown to act at telomeres that lack POT1 (refs. 25,55). Finally, it is curious that the blunt leading-end telomeres formed in the absence of Apollo are not processed by c-NHEJ, despite their interaction with DNA-PK. Perhaps TRF2 inhibits c-NHEJ in S and G2 through Rap1 (ref. 56) or by binding to Ku70/80 (ref. 57). Such S- or G2-specific mechanisms of c-NHEJ repression may also account for the lack of c-NHEJ repair of telomere-internal DSBs created in S or G2 (ref. 24).

## Online content

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

## Methods

### Cell lines and cell treatments

SV40LT $Apollo^{F/F}$, $Trf2^{F/F}Rosa26Cre-ER^{T1}$, $Trf2^{F/F}53bp1^{-/-}Rosa 26Cre-ER^{T1}$, $Trf2^{F/F}Nbs1^{F/+}$ and $Trf2^{F/F}Nbs1^{F/-}$ MEFs have been previously described[17,21,58]. $Apollo^{F/F}Dnapkcs^{-/-}$, $Apollo^{F/F}Ku70^{-/-}$, $Apollo^{F/F}Dnapkcs^{-/-}Ku70^{-/-}$ and $Apollo^{F/F}Lig4^{-/-}$ MEFs were obtained by intercrosses between the respective single-mutant mice[59–61]. To generate the $Ku70^{F/F}$ mouse, the Ku70 gene ($Xrcc6$, chromosome 15) was modified by gene targeting. Male ES cells $Xrcc6^{tm1a(KOMP)Mbp}$ were obtained from the Knockout Mouse Programme (KOMP) and used to derive heterozygous mice. The LacZ/Neo insert was removed by crossing to flippase mouse, resulting in the floxed allele ($Ku70^{F}$). Standard crosses of $Ku70^{F/+}$ mice were used to derive $Ku70^{F/F}$ MEFs. Mice ($Mus musculus musculus$, strain mixed C57BL/6 and 129) were housed and cared for under the Rockefeller University AIACUC protocol 22030-H at the Rockefeller University's Comparative Bioscience Center, which provides animal care according to NIH guidelines. All MEFs were isolated from embryonic day 12.5 (E12.5) or E13.5 embryos, immortalized at passage 2 with pBabeSV40LargeT (a gift from G. Hannon) and cultured in Dulbecco's Modified Eagle Medium (DMEM) (Cellgro) supplemented with 15% fetal bovine serum (FBS) (Gibco), non-essential amino acids (Gibco), L-glutamine (Gibco), penicillin–streptomycin (Gibco) and 50 μM β-mercaptoethanol (Sigma). Genotyping was performed by Transnetyx using real-time PCR. 293T/17 (HEK 293T/17) (ATCC CRL-11268) and Phoenix ECO cells (ATCC CRL-3214) were cultured in Dulbecco's Modified Eagle Medium (DMEM) (Corning) supplemented with 10% HyClone Calf Serum (Cytiva), non-essential amino acids (Gibco), L-glutamine (Gibco) and penicillin–streptomycin (Gibco). PARP inhibitor (PARPi) (Olaparib, AZD2281/KU-59436, BioVision) was dissolved in DMSO and added at a final concentration of 2 μM for 24 h. Rosa26Cre-ER$^{T1}$ was induced by incubation with 1 μM 4-OHT for 24 h.

### Viral gene delivery

For retroviral or lentiviral transduction, a total of 20 μg plasmid DNA was transfected into Phoenix Eco or 293T cells, respectively, using CaPO$_4$ precipitation. The viral supernatant was filtered through a 0.45-μm filter, supplemented with 4 μg ml$^{-1}$ polybrene and used for the transduction of target cells. Cre was induced with three infections per day (6–12-h intervals) over 2 d. pMMP Hit & Run Cre retrovirus was produced in Phoenix ECO cells or by adding 1 μM 4-OHT (4-OHT; Sigma H7904) to the medium. Time point 0 was set 12 h after the first Hit & Run Cre infection or at the time of 4-OHT addition. Lentiviral particles containing the shRNAs targeting ligase 3 (target: 5′-CCAGACTTCAAACGTCTCAAA-3′; TRCN0000070978, Sigma), or DNA polymerase theta (target: 5′-CGGTCCAACAAGGAAGGATTT-3′; TRCN0000120312, Sigma) in a pLKO.1 vector (Open Biosystems) were produced in 293T cells and introduced into target MEFs with 3 infections per day (6–12-h intervals) over 2 d. Infected cells were then transfected with Hit & Run Cre and selected for over 3 d in 2–4 μM puromycin before collection. Lentiviral particles containing lentiCRISPR v2 with or without sgRNA against Nbs1 (5′-GAGAATTACTGTAATCCGCA-3′, designed on Benchling (Biology Software) (2021)) were produced in 293T cells and introduced into target MEFs with six infections (4–12-h intervals) over 2 d after 4 infections with Hit & Run Cre (6–12-h intervals) over 2 d. Infected cells were selected for 2 d in 2–4 μM puromycin before collection. Retroviral particles containing the shRNAs for Exo I (target: 5′-GCATTTGGCACAAGAATTA-3′) in pSuperior vector were produced in Phoenix ECO cells and introduced into target MEFs with three infections per day (6–12-h intervals) over 2 d before selection in puromycin.

### Immunoblotting

Cells were lysed in 2× Laemmli buffer at $5 \times 10^3$ cells μl$^{-1}$, and the lysate was denatured for 10 min at 95 °C before shearing with an insulin needle. Lysate equivalent to $1 \times 10^5$ cells was resolved using SDS–PAGE and transferred to a nitrocellulose membrane. Western blot was performed with 5% milk in PBS containing 0.1% (vol/vol) Tween-20 (PBS-T) using antibodies to the following: β-actin (no. 3700; Cell Signal; 1:1,000); Chk2 (BD 611570; BD Biosciences; 1:800); DNA-PKcs (SC-1552; Santa Cruz Biotechnology; 1:200); Ku70 (sc-17789 or sc-1487; Santa Cruz Biotechnology; 1:200); Lig3 (SC-135883; Santa Cruz Biotechnology; 1:1,000); Nbs1 (ab175800; Abcam; 1:1,000); TRF2 (no. 13136; Cell Signal; 1:500); γ-tubulin (GTU-88; GeneTex; 1:1,000); and secondary HRP-conjugated anti-mouse/anti-rabbit IgG (Cytiva).

Signals were detected according to the manufacturer's instructions using chemiluminescence western blotting detection reagents (Cytiva) and either BioMax MR film (Kodak) or ChemiDoc (Bio-Rad).

### Chromosome orientation fluorescence in situ hybridization and immunofluorescence–fluorescence in situ hybridization

CO-FISH and immunofluorescence–fluorescence in situ hybridization (IF–FISH) were performed as previously described[62], with minor changes. Briefly, for CO-FISH, cells were labeled with BrdU (7.5 μM) or BrdC (2.5 μM) for 16 h and treated with 0.2 μg ml$^{-1}$ colcemid (Biowest) for 1 or 2 h before collection by trypsinization. Collected cells were incubated in the hypotonic solution 0.055M KCl at 37 °C for 30 min before fixation in methanol:acetic acid (3:1) overnight at 4 °C. Cells were dropped onto glass slides and allowed to dry overnight. Slides were then rehydrated with PBS, treated with 0.5 mg ml$^{-1}$ RNase A (R5000; Sigma) in PBS for 10 min at 37 °C, stained with 1 μg ml$^{-1}$ Hoechst 33258 (B2883; Sigma) in 2×SSC for 15 min and exposed to 5.4 × 10$^3$ J m$^{-2}$ 365-nm UV light (Stratalinker 1800 UV irradiator). After digestion with 600 U Exonuclease III (M1815, Promega) for 30 min, slides were dehydrated through an ethanol series of 70%, 95% and 100% and allowed to dry. Staining was performed in hybridization solution (70% formamide, 1 mg ml$^{-1}$ blocking reagent (1109617601, Roche) and 10 mM Tris-HCl pH 7.2) with PNA probes from PNA Bio: Cy3-OO-(CCCTAA)$_3$ and Alexa-Fluor-488-OO-(TTAGGG)$_3$, or Alexa-Fluor-647-OO-(CCCTAA)$_3$, and Cy3-OO-(TTAGGG)$_3$. Washes were performed twice in washing solution no. 1 (70% formamide; 0.1% BSA; 10 mM Tris-HCl, pH 7.2) and three times in washing solution no. 2 (0.08% Tween-20; 0.15 M NaCl; 0.1 M Tris-HCl, pH 7.2), or in PBS. DAPI (D1306, Invitrogen) was added to the second wash to stain DNA. Slides were left to dry and mounted with Prolong Gold Antifade (P36934, Fisher) embedding medium.

For IF-FISH, MEFs were grown on coverslips precoated with poly-D-lysine (A3890401, Gibco) for 1–2 d. Cells were rinsed in cold PBS and pre-extracted using cold Triton X-100 buffer (0.1% Triton X-100; 20 mM Hepes-KOH, pH 7.9; 50 mM NaCl; 3 mM MgCl$_2$; 300 mM sucrose) for 20 min on ice, followed by two washes in 1× PBS at RT, before fixation for 10 min at RT with 3% paraformaldehyde and 2% sucrose. Cells were permeabilized for 15 min with 0.1% Triton X-100 buffer before blocking and staining in blocking solution (1 mg ml$^{-1}$ BSA; 3% goat serum; 0.1% Triton X-100; 1 mM EDTA, pH 8, in PBS). A primary antibody to γH2AX (JBW301, Millipore; 1:1,000) and a secondary anti-mouse-Alexa-Fluor-647 antibody (A32728, Invitrogen) were incubated overnight at 4 °C or 1 h at RT, respectively. Samples were again fixed in 3% paraformaldehyde and 2% sucrose for 10 min at RT before dehydration through the ethanol series of 70%, 95% and 100% and allowed to dry. Hybridization was performed with Alexa-Fluor-488-OO-(TTAGGG)$_3$ in hybridization solution (70% formamide; 0.5% blocking reagent (1109617601, Sigma); 10 mM Tris-HCl, pH 7.2) for 10 min at 45 °C on a heat block, followed by incubation at RT for 2 h. After two washes in washing solution (70% formamide; 10 mM Tris-HCl, pH 7.2) and three in PBS, in which DAPI was added to stain the cell nuclei, coverslips were left to dry and mounted with Prolong Gold Antifade embedding medium.

Pictures were acquired on a Leica DMi8 microscope (Leica Microsystems) Hamamatsu ORCA Flash 4 sCMOS V3 camera with a ×63 1.2 numerical aperture (NA) objective or a DeltaVision RT microscope system (Applied Precision-GE Healthcare) with a PlanApo ×60 1.40 NA objective lens (Olympus America). For DeltaVision RT, acquisition was

performed at 1 × 1 binning and multiple 0.2-μm Z-stacks using SoftWoRx software; images were deconvolved, and 2D-maximum intensity projection images were obtained using SoftWoRx software. Chromatid and chromosome-type fusions were analyzed using Fiji (1.0) software[63] after arbitrary assignment of red for both (TTAGGG)$_3$-probes and green for both (CCCTAA)$_3$-probes.

Semi-automated analysis and quantification of colocalization was performed using CellProfiler[64] with the following pipeline: image cropping to remove edge artifacts due to deconvolution; channel intensity rescaling to cover the full histogram range value; 'speckle features enhancement' to increase detection sensitivity and remove background or artifact aggregates; 'channel-wise primary objects identification' to detect individual nuclei and individual foci; and correlation of the foci coordinates in the different channels and with the respective nuclei to define colocalization events. Nuclei with fewer than 10 detected PNA foci were discarded.

### RNA extraction and qRT–PCR

RNA was extracted from $1 \times 10^6$ cells using the RNeasy Mini Kit (Qiagen). Then, 500 ng of RNA was reverse transcribed using the First Strand cDNA Synthesis Kit (Thermo Scientific). SYBR Green PCR Master Mix (Applied Biosystems) was used for quantitative PCR. Primers used were:

β-actin-forward (F): 5′-TTCTACAATGAGCTGCGTGTGG-3′ (ref. [65])
β-actin-reverse (R): 5′-ATGGCTGGGGTGTTGAAGGT-3′ (ref. [65])
PolQ-F: 5′-GCTACCTCCAGAGTCTGTTTCAG-3′
PolQ-R: 5′-ATCCACGACCACCATTCCTAAC-3′

### In-gel analysis of single-stranded telomeric DNA

Mouse telomeric DNA was analyzed on Clamped homogenous electric field (CHEF) gels, as described previously[62]. Briefly, cells were collected by trypsinization, resuspended in PBS, mixed with 2% agarose (1:1 ratio) at 50 °C and cast in a plug mold $0.7 \times 10^6 - 1 \times 10^6$ cells per plug. Plugs were digested overnight at 50 °C in 1 mg ml$^{-1}$ proteinase K (03115879001; Roche) in digest buffer (100 mM EDTA, 0.2% sodium deoxycholate and 1% sodium lauryl sarcosine) and washed five times in TE. DNA was digested overnight at 37 °C by 60 U MboI (no. R0147; New England BioLabs). Plugs were then washed in TE, equilibrated in 0.5× TBE and loaded on a 1% agarose/0.5× TBE gel. DNA was resolved by a CHEF-DRII PFGE apparatus (Bio-Rad) for 20 h, with the following settings: initial pulse, 5 s; final pulse, 5 s; 6 V cm$^{-1}$ at 14 °C. Gel was dried and hybridized overnight at 50 °C with a [γ-$^{32}$P]ATP end-labeled TelC (AACCCT)$_4$ probe in Church mix (0.5 M sodium phosphate buffer pH 7.2, 1 mM EDTA, 7% SDS, 1% BSA). After three washes in 4× SSC and one in 4× SSC/0.1% SDS at 55 °C, the gel was exposed for 1 or 2 d, and the single-stranded telomere signal was captured by Typhoon PhosphoImager. For the acquisition of the total telomere signal, the gel was denatured with 1.5 M NaCl/0.5 M NaOH for 1 h, neutralized with two washes of 1 h each in 0.5 M Tris-HCl pH 7.0/3 M NaCl, pre-hybridized for 30 min at 55 °C in Church mix and hybridized overnight at 55 °C with the same probe. The denatured gel was washed and exposed as described before. Quantification of the signals in each lane was done using ImageQuant software. After subtraction of the background, the single-stranded signal was normalized to the total telomeric DNA signal in the same lane. The indicated control value was set to 1, and all the other values were given as a percentage of it.

### Generation and expression of MYC-TRF2 mutant alleles

PCR was used to generate ΔiDDR mutant alleles of MYC-tagged mTRF2 (MYC-TRF2) in pLPC retroviral vector using previously published constructs[7] as templates and the following primers:

TRF2$^{ΔiDDR}$-F: 5′-GTTCAGGCACCAGGTGAAGACAG-3′
TRF2$^{ΔiDDR}$-R: 5′-TGCTTTGGGCTTCTTCTCCCCCG-3′

A total of 4 infections at 6–12-h intervals were performed before selection for 2–5 d in 2–4 μM puromycin.

### Protein purification and nuclease assay

**TRF2.** Human TRF2 and TRF2-ΔiDDR were cloned into a modified pFastbac vector with a His$_6$-MBP tag and a 3C protease cleavage site. Proteins were expressed in insect cells grown for 72 h after infection with baculovirus, collected, frozen in liquid nitrogen and stored at −80 °C. Cell pellets were thawed and homogenized in lysis buffer (40 mM Tris pH 8, 500 mM NaCl, 0.5 mM TCEP, 10% glycerol, 0.1% Tween-20, 1 mM PMSF, protease inhibitor cocktail (Roche)). Cells were pelleted at 18,000g and incubated with 1 ml Ni-NTA resin for 1 h. The resin was washed with buffer A (40 mM Tris pH 8, 100 mM NaCl, 0.5 mM TCEP, 10% glycerol) and proteins were eluted in buffer A and 200 mM imidazole. Proteins were incubated with 3C protease overnight at 8 °C and injected on a Hi-Trap Heparin column (Cytiva). After being washed extensively with buffer A, proteins were eluted in buffer B (40 mM Tris pH 8, 1 M NaCl, 0.5 mM TCEP, 10% glycerol). The most concentrated fractions were injected into a Superose 6 column (Cytiva) equilibrated in buffer A. Fractions were analyzed on SDS–PAGE for purity, pooled, concentrated to 1 mg ml$^{-1}$, aliquoted and flash-frozen in liquid nitrogen.

**MRN, CtIP and DNA-PK.** MRN, CtIP and DNA-PK (Ku70/80 and DNA-PKcs) were purified as described previously[30]. CtIP purified from insect cells is phosphorylated.

**Endonuclease assay.** The MRN–CtIP endonuclease assay was performed as described[30], with the following modifications: MRN was preincubated with the noted concentrations of TRF2 or TRF2-ΔiDDR for 10 min on ice before addition of a separately prepared mixture containing DNA-PK, 5′ $^{32}$P-labeled substrate DNA and NU7441. Endonuclease activity was initiated by addition of the remaining components, and CtIP and reaction products were assessed after 1 h at 37 °C on polyacrylamide gels containing 12% polyacrylamide, 20% formamide and 6 M urea, and were imaged by phosphorimager.

**Exonuclease assay.** MRN exonuclease assay was performed essentially as described[42,66]. Briefly, PC1253C (5′-AACGTCATAGACGATTACATT GCTAGGACATCTTTGCCCACGTTGACCCA-3′) was labeled at the 3′ end with alpha-ATP by TdT for 1 h at 37 °C, heat inactivated at 75 °C for 20 min and purified on a G-25 spin column. Labeled oligonucleotide (100 nM final) was annealed to 2× concentration (200 nM final) of PC1253B (5′-TGGGTCAACGTGGGCAAAGATGTCCTAGCAATGTAATCGTCTAT GACGTT-3′) with a 3′ Biotin-TEG label. Labeled substrate (1 nM final) was preincubated with 15 nM streptavidin in reaction buffer (16 mM Tris pH 8, 40 mM NaCl, 4% glycerol, 0.2 mM TCEP, 5 mM MgCl$_2$, 1 mM MnCl$_2$, 1 mM ATP, 0.25 mg ml$^{-1}$ BSA). Mre11–Rad50 (120 nM final) was preincubated with 2.5× Nbs1 (300 nM final) and TRF2 at the indicated concentration before being added to the reaction mix. The reaction was incubated at 37 °C for 2 h before stop buffer (16 mM EDTA, 0.3 mg ml$^{-1}$ Proteinase K and 0.3% SDS final) was added, followed by incubation at 50 °C for 30 min. Products were resolved on a 15% TBE-Urea gel (Thermo) and visualized on a Phosphoimager. Four independent replicates were performed and quantified using GraphPad Prism 9.

### AlphaFold-Multimer and evolutionary and structural analysis

AlphaFold-Multimer (v2.1.0)[39] was run locally on a GPU workstation using default parameters. A custom script (modified from AlphaFold Colab) was used to extract PAE and pLDDT information from the resulting pickle files, and structures were analyzed in PyMol (Schrodinger) and ChimeraX (UCSF). TRF2, TRF, CtIP and Rad50 protein sequences were obtained from PSI-BLAST searches of the non-redundant protein sequences (nr) database against human sequences (https://blast.ncbi.nlm.nih.gov/). Alignment of the sequences was performed using MUSCLE in SnapGene and formatted in Jalview. The CtIP alignment was visualized using the NCBI MSA Viewer (v1.22.2) and colored according to 'conservation.'

## Quantification and statistical analysis

Quantification and statistical analysis were performed using Microsoft Excel and GraphPad, respectively, over three or more independent experiments, as indicated. For each CO-FISH analysis, at least ten metaphases per condition per experiment were scored. For in-gel analysis of single-stranded telomeric DNA, the normalized overhang signals were expressed for each cell line independently. Significance was assessed by calculating the $P$ value using Kruskal–Wallis one-way ANOVA without Gaussian distribution assumption (to compare telomere fusions), unpaired $t$-test without Gaussian distribution assumption (to compare mRNA expression and telomere overhang signal in a single cell line) and two-way ANOVA without Gaussian distribution assumption (to compare telomere overhang signal from more than one cell line). $P$ values $\leq 0.05$ were considered statistically significant.

## Reporting summary

Further information on research design is available in the Nature Portfolio Reporting Summary linked to this article.

## Data availability

All data generated or analyzed during this study are included in this article. Source data are provided with this paper. All raw images are available on FigShare (https://doi.org/10.6084/m9.figshare.23650509) or from the corresponding authors upon reasonable request. Source data are provided with this paper.

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

## Acknowledgements

We are extremely grateful to D. White for mouse husbandry. We thank Y. Zhang for the preliminary experiments with PARPi, A. Panza for help with CellProfiler, and T. Walz and S. Cai for help with AlphaFold-Multimer. This work was supported by grants from Cancerfonden (CAN 2018/493 to F.L.), Vetenskapsrådet (ÄR-MH 2018-03215 to F.L.) and the NIH (R35 CA210036 and AG016642 to T.d.L.). F.L. is a Wallenberg Molecular Medicine fellow and receives financial support from the Knut and Alice Wallenberg Foundation. B.T. was partially supported by the Lions forskningsfond (LiU-2022-01245). T.T.P. and C.K.V. were supported by NIH (R01GM138548). The funders had no role in study design, data collection and analysis, decision to publish or preparation of the manuscript.

## Author contributions

L.R.M. performed the cell biology, biochemistry and structural analysis of iDDR and MRN–CtIP interaction. B.T. and F.L. performed all the analyses of Apollo deletion and characterized the role of iDDR in protecting unprocessed telomeres. B.T., A.M.M. and L.R.M. performed the overhang analysis of Apollo- and alt-EJ-decifient cells. C.K.V. performed the the inhibtion of MRN–CtIP assay with supervision from T.T.P. K.T. generated and performed all the experiments with $Ku70^{F/F}$ MEFs. P.W. and F.L. generated all the other MEFs. P.W. discovered the fusion phenotype of Apollo- and Lig4-deficient MEFs. A.M.M. discovered the role of Nbs1 in preventing fusions in the absence of Apollo and DNA-PK. F.L., T.d.L. and L.R.M. conceptualized the study with B.T. F.L. and T.d.L. procured funding and wrote the paper, with contributions from all the other authors.

## Funding

## Competing interests

T.d.L. is a member of the scientific advisory board of Calico, San Francisco, CA, USA. The remaining authors declare no competing interests.

## Additional information

**Extended data** is available for this paper at https://doi.org/10.1038/s41594-023-01072-x.

**Correspondence and requests for materials** should be addressed to Titia de Lange or Francisca Lottersberger.

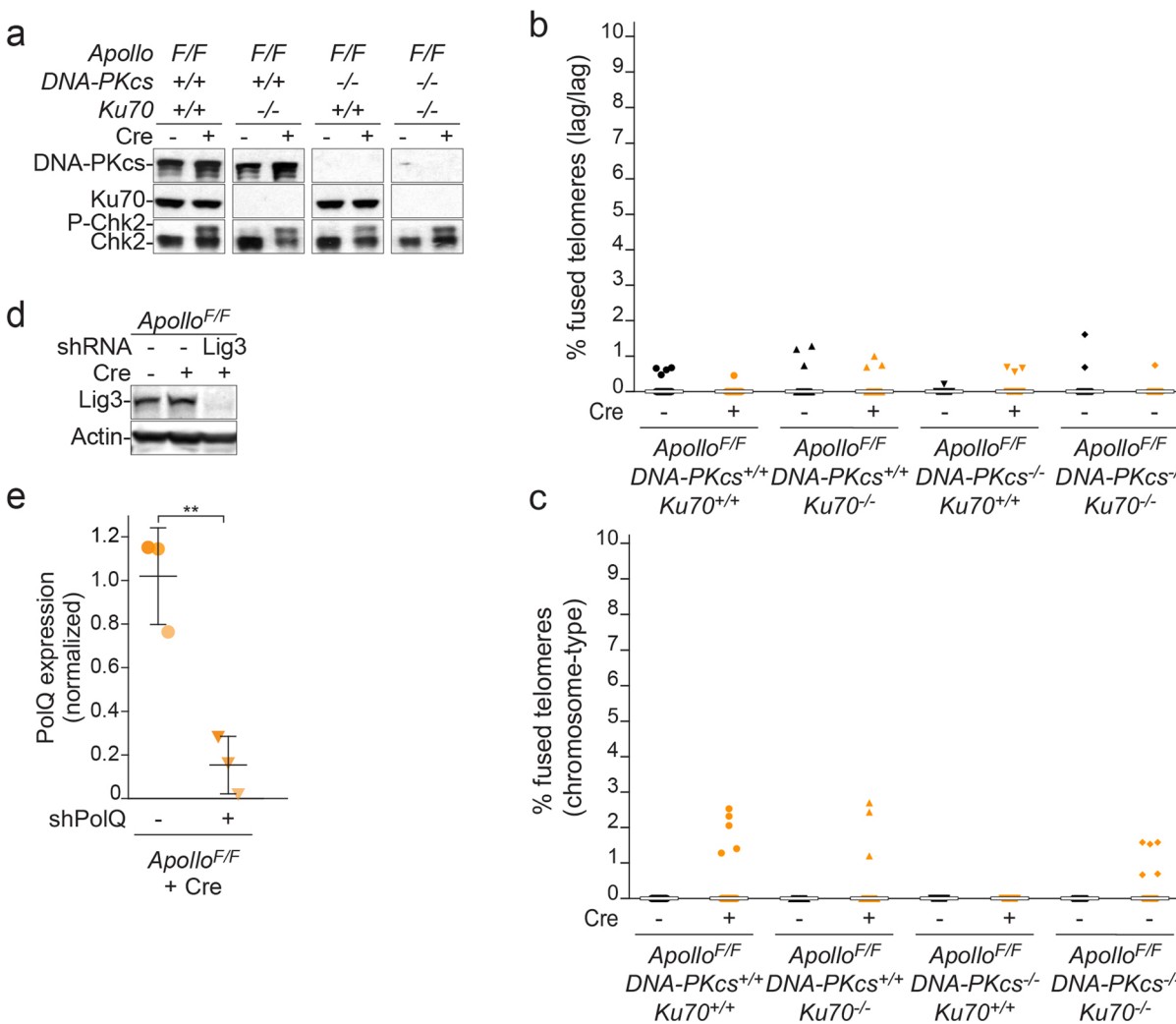

**Extended Data Fig. 1 | DNA-PK does not affect Chk2 phosphorylation after Apollo deletion. a**) Immunoblots for DNA-PKcs, Ku70, and phosphorylated Chk2 in SV40LT-immortalized $Apollo^{F/F}$, $Apollo^{F/F} Ku70^{-/-}$, $Apollo^{F/F} DNA\text{-}PKcs^{-/-}$ or $Apollo^{F/F} Ku70^{-/-} DNA\text{-}PKcs^{-/-}$ MEFs, without any further treatment or 96 h after transduction with Hit & Run Cre, as analyzed in Figs. 1a,b and 2a,b. (**b**) and (**c**) Quantification of telomere fusions as shown in Fig. 1a aggregated for chromatid-type involving two lagging-end telomeres (lag/lag) (b) or chromosome-type fusions (chromosome) (c). Bars represent the median over 10 metaphases for three independent experiments (30 metaphases in total). (**d**) Immunoblots for

Lig3 in SV40LT-immortalized $Apollo^{F/F}$ MEFs after transduction with the empty vector or the shRNA against Lig3 and/or 108 h after treatment with Hit & Run Cre as analyzed in Fig. 1g,h. (**e**) PolQ mRNA expression normalized to β-actin in SV40LT-immortalized $Apollo^{F/F}$ MEFs after transduction with the empty vector or the shRNA against PolQ and 108 h after treatment with Hit & Run Cre, as analyzed in Fig. 1g,h. Values were obtained from three independent experiments and normalized to the empty vectors, with means and SD. Statistical analysis by unpaired t-test with Welch correction.

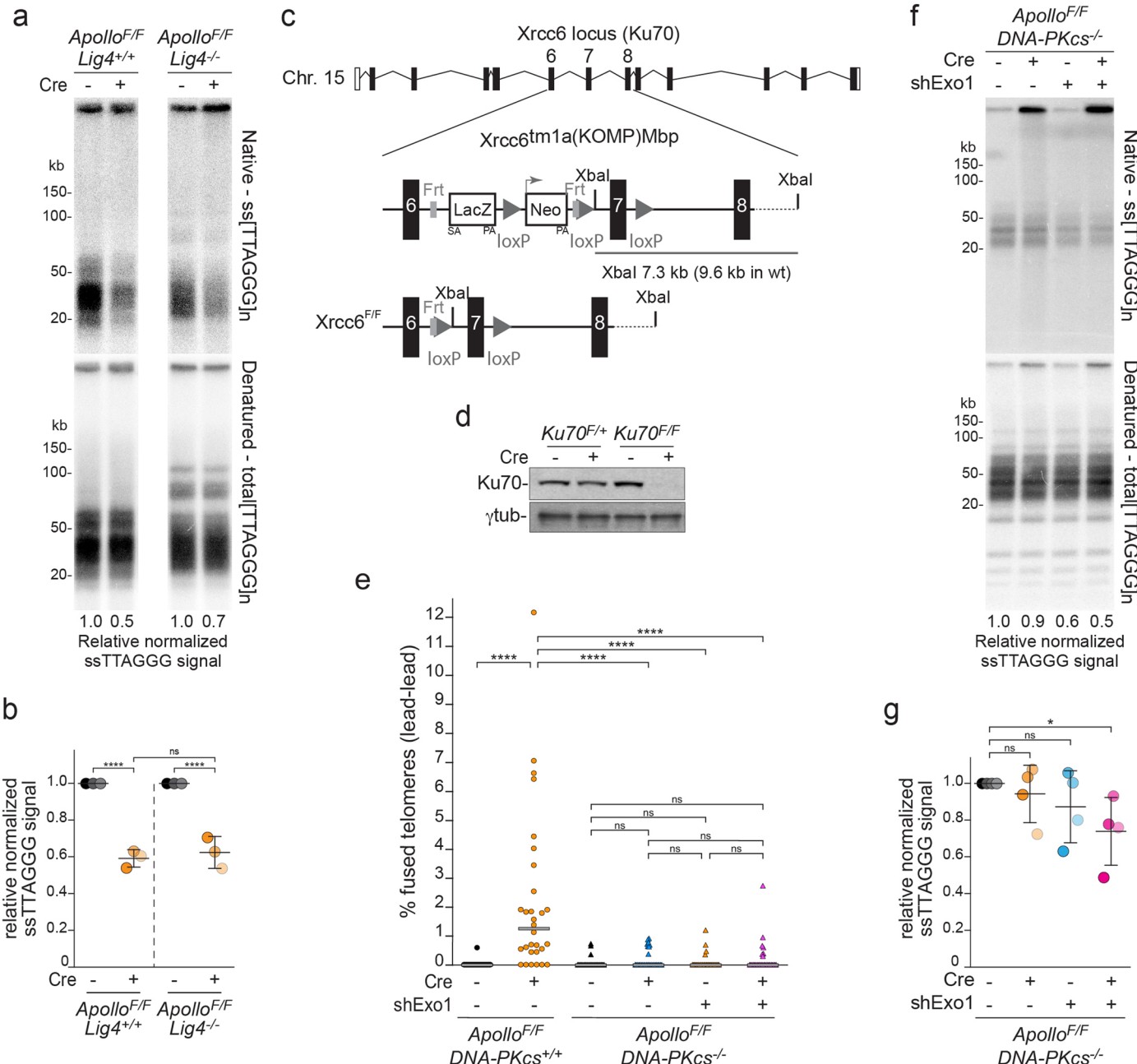

**Extended Data Fig. 2 | Exo1 promotes Apollo-independent processing of telomere overhang in absence of DNA-PKcs.** (**a**), (**b**) Telomeric overhang assay and quantification on SV40-LT-immortalized *Apollo*^F/F *Lig4*^+/+ and *Apollo*^F/F *Lig4*^−/− MEFs 96 h after Cre-mediated deletion of endogenous Apollo for three independent experiments. Statistical analysis by two-way ANOVA. (**c**) Targeting of the mouse XRCC6/KU70 locus. The Xrcc6 genomic locus, the KOMP-derived targeted allele with the LacZ/Neo insert and the floxed allele are indicated. The LoxP sites are represented as triangles. (**d**) Immunoblots for mouse Ku70 in SV40-LT-immortalized *Ku70*^F/+ or *Ku70*^F/F without any treatment or 108 h after

viral transduction with Hit & Run Cre as analyzed in Fig. 2e,f. (**e**) Quantification of leading end telomere fusions in *Apollo*^F/F *DNA-PKcs*^−/− MEFs 108 h after Cre-mediated deletion of endogenous Apollo and/or depletion of Exo1 for two independent experiments. Bars represent median of 20 metaphases (10 per experiment). Statistical analysis by Kruskal-Wallis one-way ANOVA for multiple comparisons. (**f**), (**g**) Telomeric overhang assay and quantification on SV40-LT-immortalized *Apollo*^F/F *DNA-PKcs*^−/− MEFs 108 h after Cre-mediated deletion of endogenous Apollo and/or depletion of Exo1 for four independent experiments. Statistical analysis by unpaired t-test.

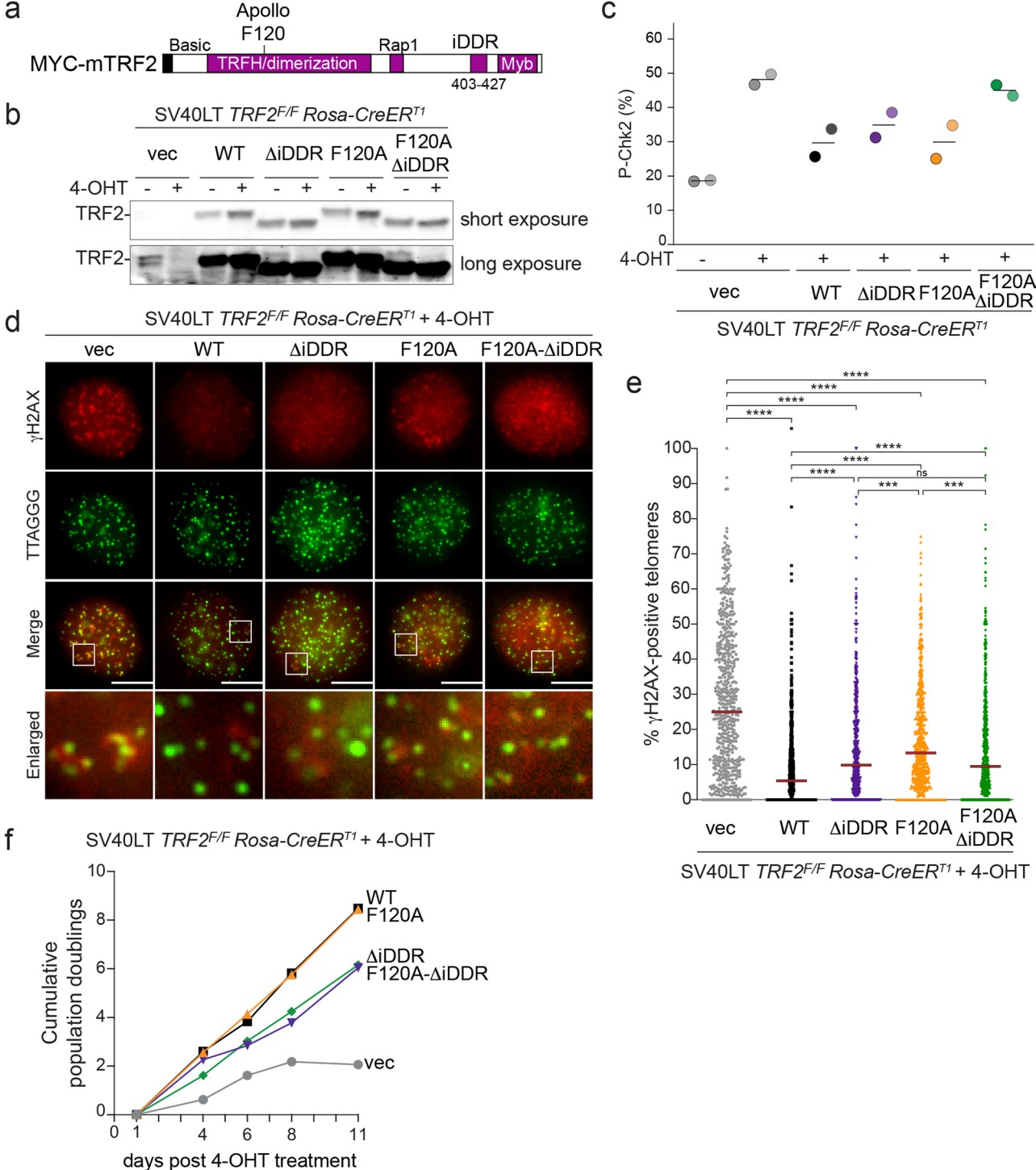

**Extended Data Fig. 3 | Expression of *MYC-TRF2* alleles. (a)** Schematic of MYC-tagged mouse TRF2 with Basic, Telomeric Repeat Factors Homology (TRH), Hinge, and Myb domains. Phenylalanine 120 (F120) required for the interaction with Apollo and the iDDR region are highlighted. **(b)** Lower and higher exposure for the immunoblot of endogenous and exogenous TRF2 as shown in Fig. 4a. **(c)** Quantification of the percentage of phosphorylated Chk2 versus total Chk2 as shown in Fig. 4a for n = 2 independent experiments. Bars indicate the average. **(d)** IF-FISH of SV40LT-immortalized *Trf2*^F/F *RsCre-ER*^T1 MEFs expressing the empty vector (EV) or the indicated *MYC-Trf2* alleles 73 h after 4-OHT-mediated deletion of endogenous TRF2. TIFs are detected by immunofluorescence with antibodies

for γ-H2AX (red) and the Telomeres-specific probe Alexa488-OO-(TTAGG)3 (green). Bars represent 10 μm. **(e)** Quantification of the percentage of TIFs as in (d). Median bars from 600 nuclei over n = 4 independent experiments (150 nuclei per experiment per conditions). Statistical analysis by Kruskal-Wallis Anova test for multiple comparisons. **(f)** Growth curve showing cumulative population doublings in SV40LT-immortalized *Trf2*^F/F *RsCre-ER*^T1 MEFs expressing the empty vector (EV), in grey, or the indicated *MYC-Trf2* alleles: *WT* in black, *ΔiDDR* in green, *F120A* in orange or *F120AΔiDDR* in purple. 4-OHT was added at time 0. One representative experiment is shown.

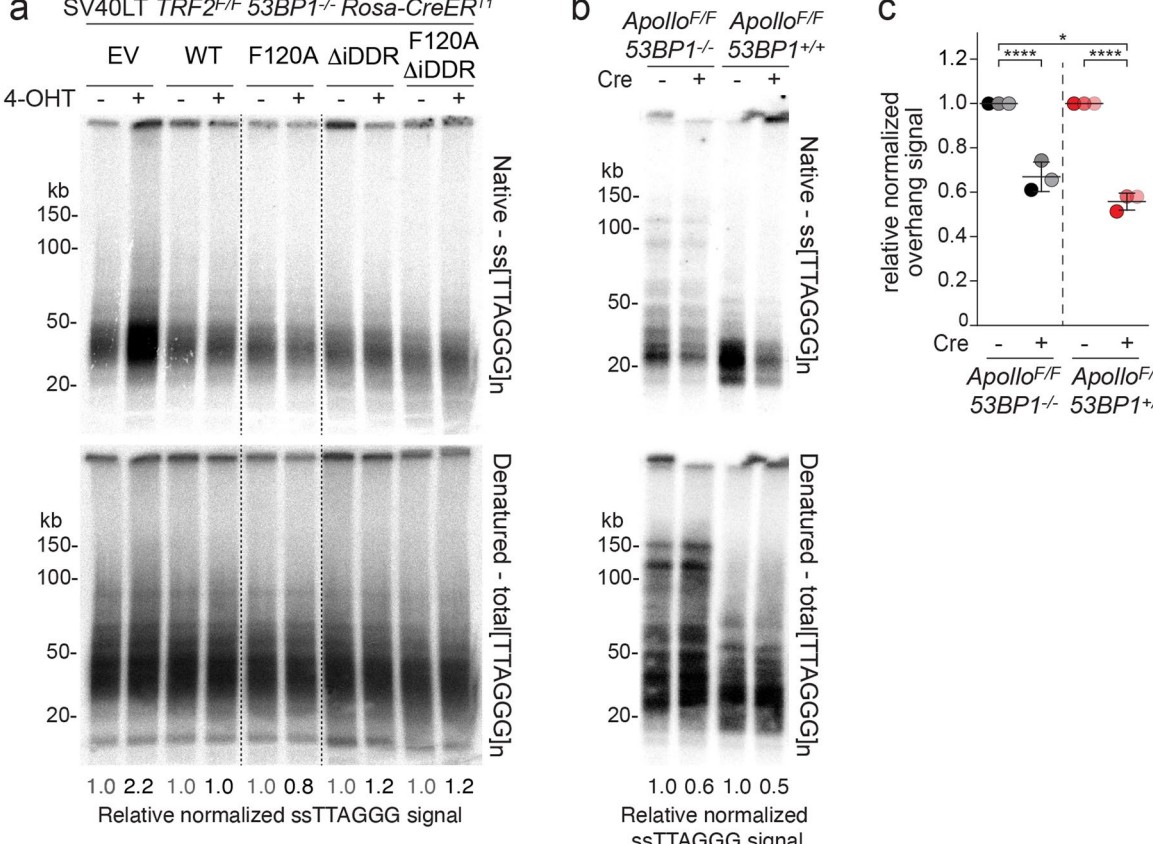

**Extended Data Fig. 4 | The iDDR prevents Apollo-independent nucleolytic processing of leading-end telomeres independently from 53BP1. (a)** Telomeric overhang assay and quantification from one representative experiment in SV40LT-immortalized *Trf2*^F/F *53BP1*^-/- *RsCre-ER*^T1 MEFs expressing the indicated *MYC-Trf2* alleles at 96 h after 4-OH tamoxifen (4-OHT)-mediated deletion of endogenous TRF2. For each allele, the normalized no Cre value was

set to 1, and the + Cre value was given relative to it. **(b)** and **(c)** Telomeric overhang assay and quantification of *Apollo*^F/F and *Apollo*^F/F *53BP1*^−/− MEFs 96 h after Hit & Run Cre-mediated deletion of Apollo as described in (a) and (b), with means and SDs across three independent experiments. Statistical analysis by two-way ANOVA.

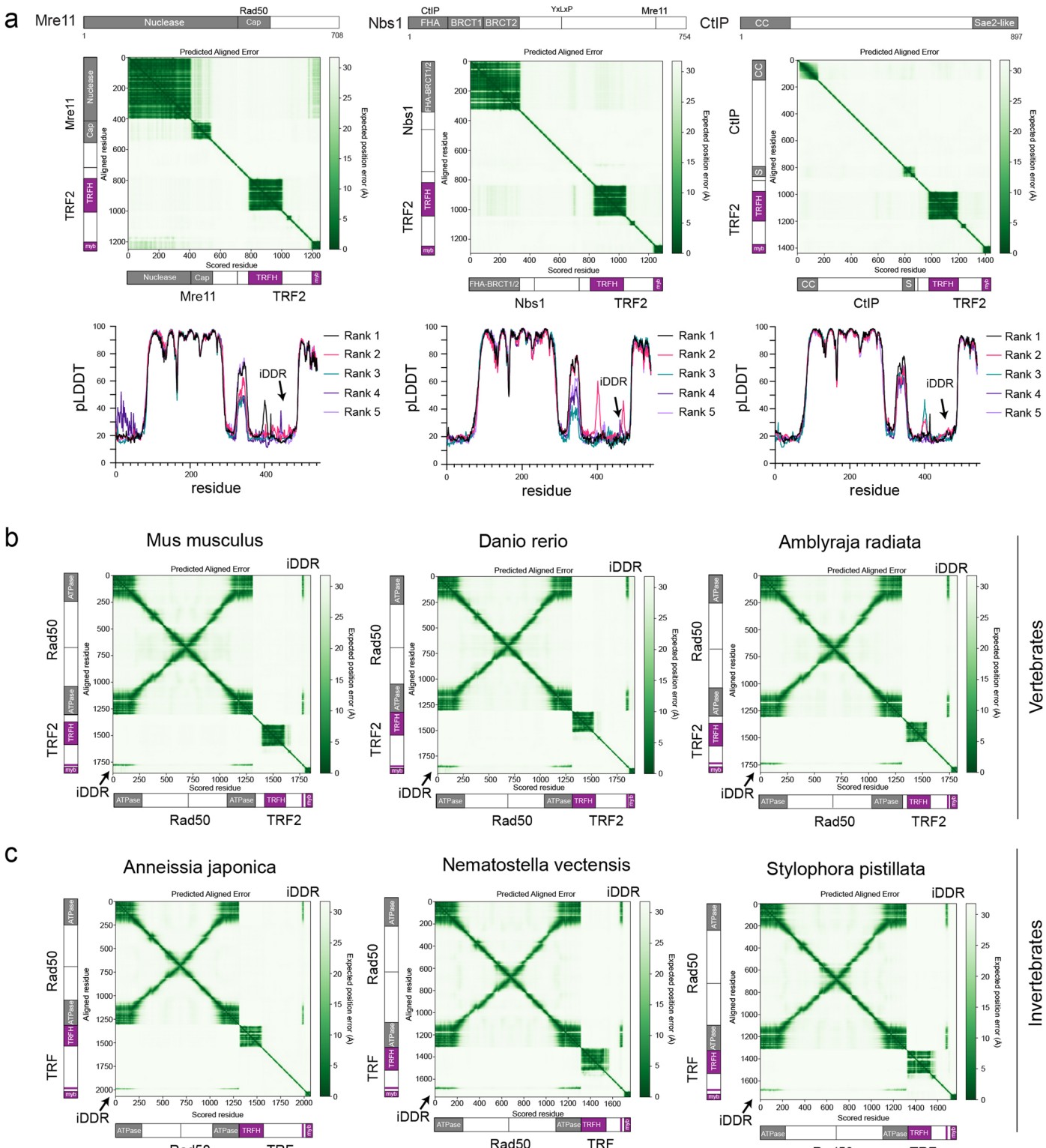

**Extended Data Fig. 5 | The iDDR is predicted to interact with Rad50 in several metazoan. (a)** Representative Predicted Aligned Error (PAE; top) of TRF2 with Mre11 (left), Nbs1 (middle), and CtIP (right) from AlphaFold-Multimer models and predicted local distance difference test (pLDDT; bottom) for each residue in TRF2 from five ranked models generated with default parameters. (**b**) Representative Predicted Aligned Error (PAE) for the interaction between TRF2 and Rad50 in representative vertebrates. (**c**) Representative Predicted Aligned Error (PAE) for the interaction between TRF and Rad50 in representative invertebrates.

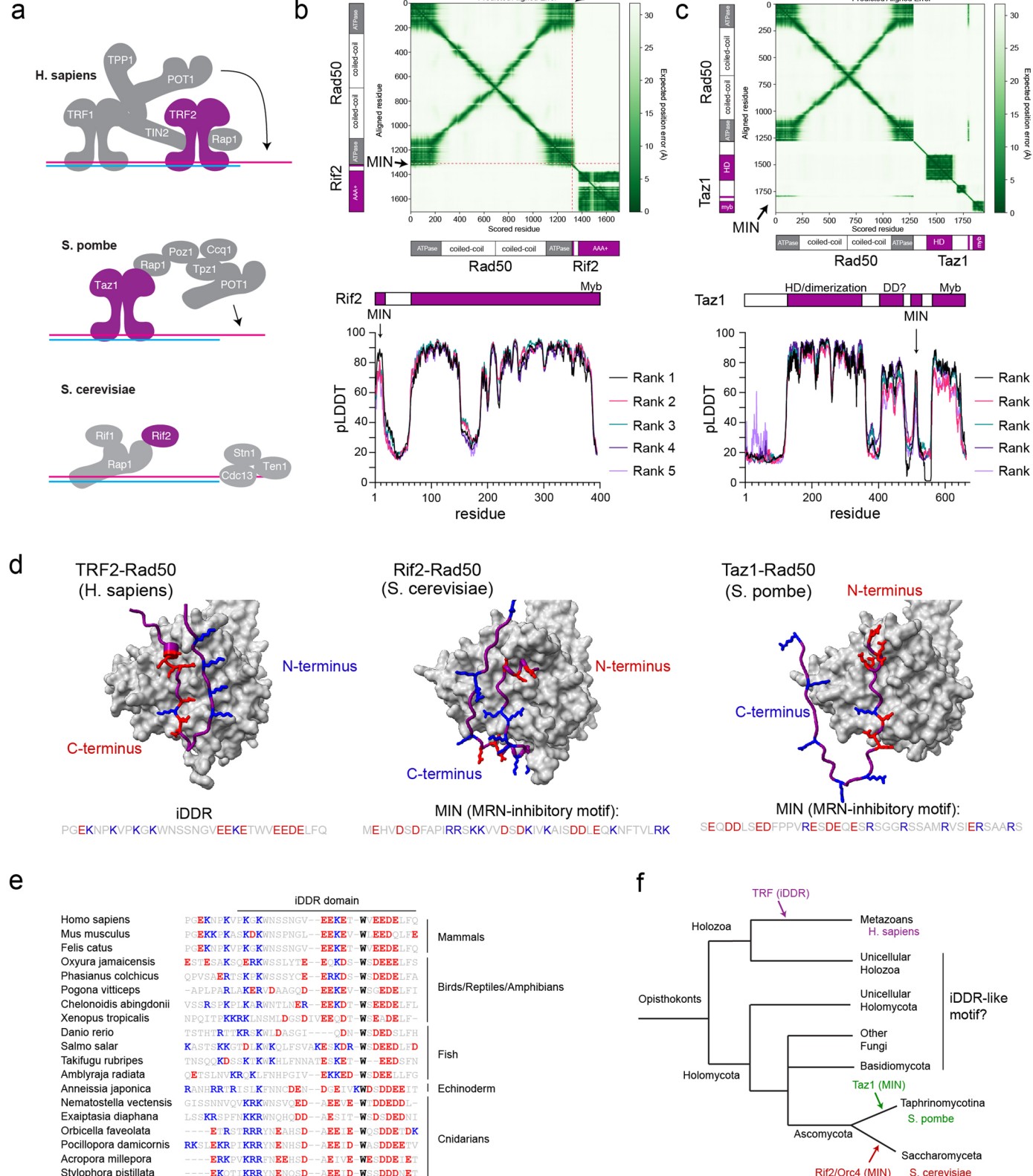

**Extended Data Fig. 6 | The iDDR of TRF2 and the MIN domains of Rif2 and Taz1 are an example of convergent evolution. (a)** Schematic for telomere binding proteins in H. sapiens, S. pombe, and S. cerevisiae with TRF2, Taz1, Rif2 highlighted in purple. **(b)** AlphaFold-Multimer Predicted Aligned Error (PAE; top; representative of five ranked models generated with default parameters) plot for S. cerevisiae Rif2 and Rad50 and predicted Local Distance Difference Test (pLDDT; bottom) plot for each residue in Rif2 from the ranked models.**(c)** AlphaFold-Multimer PAE (top; representative of five ranked models generated with default parameters) plot for S. pombe Taz1 and Rad50 and pLDDT (bottom)

plot for across Taz1 from the ranked models. **(d)** Representative AlphaFold-Multimer models of TRF2-Rad50 (left), Rif2-Rad50 (middle), and Taz1-Rad50 (right). Acidic (red) and basic (blue) residues are highlighted. **(e)** MUSCLE alignment of the iDDR domains from representative metazoans, highlighting the presence of basic residues (blue) followed by acidic residues (red). **(f)** Phylogenetic tree of Opisthokonts showing the emergence of the iDDR of TRF proteins (blue) as well as the MIN of Taz1 (green) and the MIN (or BAT) of Rif2/Orc4 (red). Whether other Opisthokonts independently evolved an iDDR-like motif is unknown. Not to scale.

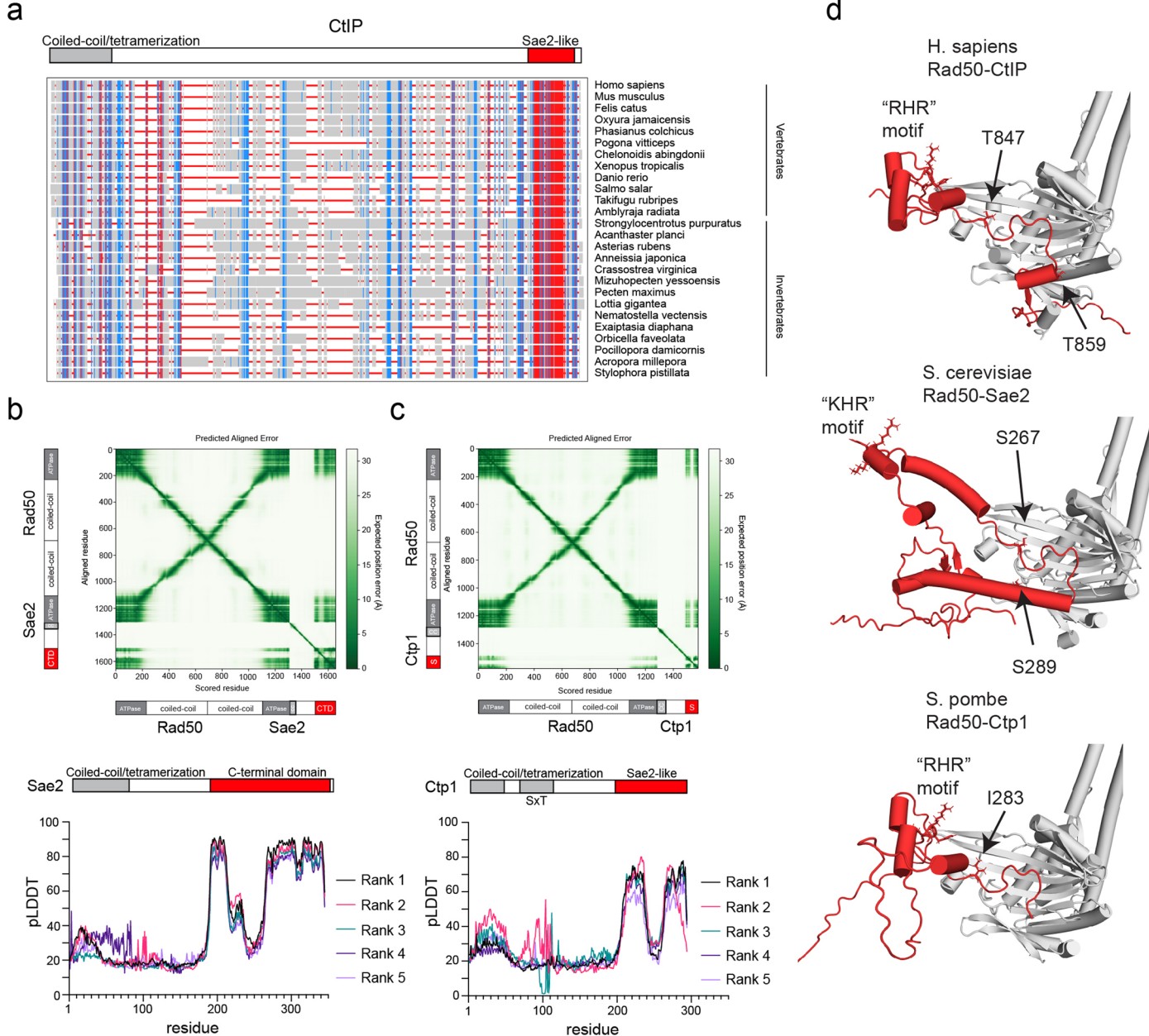

**Extended Data Fig. 7 | CtIP, Sae2, and Ctp1 are all predicted to interact with Rad50 in a similar manner.** (**a**) Multiple Sequence Alignment (MSA) of CtIP proteins from vertebrate and invertebrate species using NCBI MSA Viewer from an alignment using Multiple Sequence Comparison by Log-Expectation (MUSCLE). Vertical lines are colored by conservation where red indicates highly conserved and blue indicates lower conservation. Alignment positions with gaps are not colored. (**b**) Representative AlphaFold-Multimer models for CtIP-Rad50, Sae2-Rad50, and Ctp1-Rad50. Important CDK and ATM/Tel1 sites are indicated as well as the residue in the ATM site position on Ctp1. (**c**) AlphaFold-Multimer Predicted Aligned Error (PAE; top; representative of five ranked models generated with default parameters) plot for S. cerevisiae Sae2-Rad50 and predicted Local Distance Difference Test (pLDDT; bottom) plot across Sae2 from the ranked models. (**d**) Representative AlphaFold-Multimer models for CtIP-Rad50, Sae2-Rad50, and Ctp1-Rad50. Important CDK and ATM/Tel1 sites are indicated as well as the residue in the ATM site position on Ctp1.

# Reporting Summary

## Statistics

For all statistical analyses, confirm that the following items are present in the figure legend, table legend, main text, or Methods section.

| n/a | Confirmed | |
|---|---|---|
| ☐ | ☒ | The exact sample size ($n$) for each experimental group/condition, given as a discrete number and unit of measurement |
| ☐ | ☒ | A statement on whether measurements were taken from distinct samples or whether the same sample was measured repeatedly |
| ☐ | ☒ | The statistical test(s) used AND whether they are one- or two-sided *Only common tests should be described solely by name; describe more complex techniques in the Methods section.* |
| ☒ | ☐ | A description of all covariates tested |
| ☐ | ☒ | A description of any assumptions or corrections, such as tests of normality and adjustment for multiple comparisons |
| ☐ | ☒ | A full description of the statistical parameters including central tendency (e.g. means) or other basic estimates (e.g. regression coefficient) AND variation (e.g. standard deviation) or associated estimates of uncertainty (e.g. confidence intervals) |
| ☐ | ☒ | For null hypothesis testing, the test statistic (e.g. $F$, $t$, $r$) with confidence intervals, effect sizes, degrees of freedom and $P$ value noted *Give P values as exact values whenever suitable.* |
| ☒ | ☐ | For Bayesian analysis, information on the choice of priors and Markov chain Monte Carlo settings |
| ☒ | ☐ | For hierarchical and complex designs, identification of the appropriate level for tests and full reporting of outcomes |
| ☒ | ☐ | Estimates of effect sizes (e.g. Cohen's $d$, Pearson's $r$), indicating how they were calculated |

*Our web collection on statistics for biologists contains articles on many of the points above.*

## Software and code

Policy information about availability of computer code

| Data collection | ChemiDoc (Bio-Rad), DeltaVision RT microscope system (GE Healthcare), Leica DMi8 microscope (Leica Microsystems), Typhoon PhosphoImager (GE) |
|---|---|
| Data analysis | AlphaFold-Multimer (v2.1.0), Benchling [Biology Software] (2021), CellProfiler, Fiji (1.0), GraphPad Prism (v9 or 10.0.0), ImageQuant, Jalview (v2.11.2.7), Microsoft Excel (16.74), MUSCLE, NCBI MSA Viewer (v1.22.2), PyMol (v2.5.4), SnapGene (v4.3.11), SoftWoRX, UCSF ChimeraX (v1.2.5). |

For manuscripts utilizing custom algorithms or software that are central to the research but not yet described in published literature, software must be made available to editors and reviewers. We strongly encourage code deposition in a community repository (e.g. GitHub). See the Nature Portfolio guidelines for submitting code & software for further information.

## Data

All manuscripts must include a data availability statement. This statement should provide the following information, where applicable:

- Accession codes, unique identifiers, or web links for publicly available datasets
- A description of any restrictions on data availability
- For clinical datasets or third party data, please ensure that the statement adheres to our policy

All data generated or analyzed during this study are included in this article. Source data are provided with this paper. All other raw images are available from the corresponding authors upon reasonable request. Protein sequences were retreived from https://blast.ncbi.nlm.nih.gov/

## Research involving human participants, their data, or biological material

Policy information about studies with human participants or human data. See also policy information about sex, gender (identity/presentation), and sexual orientation and race, ethnicity and racism.

| | |
|---|---|
| Reporting on sex and gender | N/A |
| Reporting on race, ethnicity, or other socially relevant groupings | N/A |
| Population characteristics | N/A |
| Recruitment | N/A |
| Ethics oversight | N/A |

Note that full information on the approval of the study protocol must also be provided in the manuscript.

# Field-specific reporting

Please select the one below that is the best fit for your research. If you are not sure, read the appropriate sections before making your selection.

☒ Life sciences          ☐ Behavioural & social sciences          ☐ Ecological, evolutionary & environmental sciences

For a reference copy of the document with all sections, see nature.com/documents/nr-reporting-summary-flat.pdf

# Life sciences study design

All studies must disclose on these points even when the disclosure is negative.

| | |
|---|---|
| Sample size | No statistical method was used to predetermine sample size. Sample size was determined based on previous similar experiments: n=30-45 metaphases over 3-4 independent experiments for telomere fusions (Lottersberger, F., Karssemeijer, R. A., Dimitrova, N. & de Lange, T. 53BP1 and the LINC Complex Promote Microtubule-Dependent DSB Mobility and DNA Repair. Cell 163, 880–893 (2015)) and n=3-5 independent experiments for telomere overhang (Wu, P., van Overbeek, M., Rooney, S. & de Lange, T. Apollo contributes to G overhang maintenance and protects leading-end telomeres. Mol Cell 39, 606–617 (2010)). |
| Data exclusions | No data were excluded from the experiments in the study but metaphases were it was not possible to perform robust scoring (i.e. double CO-FISH staining, less than 10 chromosomes, all overlapping chromosomes) and nuclei with less than 10 telomere foci. In case of failed experiment (Cre or shRNA not working properly), all data associated with that specific experiment were not included. |
| Replication | The number of independent experiments performed is indicated in the figure legends. All sample images are representative of at least n=3 independent experiments with similar results, unless indicated. |
| Randomization | Randomization was done when possible. Culture dishes seeded at the same time with identical population were randomly chosen for the subsequent biological perturbation (Cre, shRNA, PARPi, TRF2 alleles). Pictures of metaphases and nuclei for all the cells/treatments were taken randomly by manual scanning of the slides/coverslips. |
| Blinding | Investigators were not blinded during the study. However, all samples were processed in parallel and treated identically for all the experiments. In most experiments blinded analysis is not applicable since the sample identity is readily apparent to the investigator. |

# Reporting for specific materials, systems and methods

We require information from authors about some types of materials, experimental systems and methods used in many studies. Here, indicate whether each material, system or method listed is relevant to your study. If you are not sure if a list item applies to your research, read the appropriate section before selecting a response.

## Materials & experimental systems

| n/a | Involved in the study |
|-----|------------------------|
| ☐ | ☒ Antibodies |
| ☐ | ☒ Eukaryotic cell lines |
| ☒ | ☐ Palaeontology and archaeology |
| ☐ | ☒ Animals and other organisms |
| ☒ | ☐ Clinical data |
| ☒ | ☐ Dual use research of concern |
| ☒ | ☐ Plants |

## Methods

| n/a | Involved in the study |
|-----|------------------------|
| ☒ | ☐ ChIP-seq |
| ☒ | ☐ Flow cytometry |
| ☒ | ☐ MRI-based neuroimaging |

# Antibodies

| | |
|---|---|
| Antibodies used | Immunoblot: beta-Actin (#3700; Cell Signal, 1:1000); Chk2 (BD 611570; BD Biosciences, 1:800); DNA-PKcs (SC-1552; Santa Cruz Biotechnology, 1:200); Ku70 (sc-17789 or sc-1487; Santa Cruz Biotechnology, 1:200); Lig3 (SC-135883; Santa Cruz Biotechnology, 1:1000); Nbs1 (ab175800; Abcam, 1:1000); TRF2 (#13136; Cell Signal, 1:500); gamma-Tubulin (GTU-88; GeneTex, 1:1000); and secondary anti-Mouse/anti-Rabbit IgG HRP (Cytiva).<br>Immunofluorescence: gamma-H2AX (JBW301, Millipore; 1:1000) primary antibodies, and secondary anti-mouse AlexaFluor 647 antibody (A32728, Invitrogen), |
| Validation | Antibodies against beta-Actin, Chk2, Lig3,  Nbs1, TRF2 and gamma-Tubulin were validated by the suppliers companies for reactivity against the mouse proteins by Immunoblot. Antibodies against gamma-H2AX were validated by the suppliers companies for reactivity against the mouse proteins by Immunofluorescence.<br>Antibodies against DNA-PKcs and Ku70 were validated by the suppliers companies for reactivity against the human proteins. We validated them for reactivity against the mouse proteins  by the Immunoblot in Extended Data Fig1 on genotyped MEFs. |

# Eukaryotic cell lines

Policy information about cell lines and Sex and Gender in Research

| | |
|---|---|
| Cell line source(s) | 293T/17 [HEK 293T/17] (CRL-11268) and  Phoenix ECO cells (CRL-3214) were obtained by ATCC, Rockville, MD).<br>MEFs used in this study were generated previously (Dimitrova, N. & de Lange, T. Cell cycle-dependent role of MRN at dysfunctional telomeres: ATM signaling-dependent induction of nonhomologous end joining (NHEJ) in G1 and resection-mediated inhibition of NHEJ in G2. Mol Cell Biol 29, 5552–5563 (2009); Lottersberger, F., Karssemeijer, R. A., Dimitrova, N. & de Lange, T. 53BP1 and the LINC Complex Promote Microtubule-Dependent DSB Mobility and DNA Repair. Cell 163, 880–893 (2015); Wu, P., van Overbeek, M., Rooney, S. & de Lange, T. Apollo contributes to G overhang maintenance and protects leading-end telomeres. Mol Cell 39, 606–617 (2010)) or for this study by F.L, K.T. or P.W. in T.d.L. laboratory. |
| Authentication | No authentication was performed for 293T and Phoenix ECO cells.<br>MEFs were genotyped by  Transnetyx Inc. using real-time PCR and authenticated when possible by Immunoblots (for DNA-PKcs, Ku70 and TRF2 deletion). |
| Mycoplasma contamination | All cells tested negative for Mycoplasma contamination |
| Commonly misidentified lines (See ICLAC register) | No commonly misidentified cell lines were used in this study. |

# Animals and other research organisms

Policy information about studies involving animals; ARRIVE guidelines recommended for reporting animal research, and Sex and Gender in Research

| | |
|---|---|
| Laboratory animals | Pregnant female mice were used to isolate MEFs from E12.5 embryos.<br>Species Mus musculus musculus;; Strain mixed C57BL/6 and 129 ; sex female and male; age range 2-10 months. |
| Wild animals | N/A |
| Reporting on sex | N/A |
| Field-collected samples | N/A |
| Ethics oversight | Mice were housed and cared for under the Rockefeller University AIACUC protocol 22030-H at the Rockefeller University's Comparative Bioscience Center, which provides animal care according to NIH guidelines. |

Note that full information on the approval of the study protocol must also be provided in the manuscript.

