## [Peer Review File · Nature Structural & Molecular Biology]

Peer Review Information

Manuscript Title: DNA-PK and the TRF2 iDDR inhibit MRN-initiated resection at leading-end telomeres

Corresponding author name(s): Francisca Lottesberger, Titia de Lange

Editorial Notes:

**Redactions –
unpublished data**

Reviewer Comments & Decisions:

Decision Letter, initial version:

Message: 17th Feb 2023

Dear Dr. Lottesberger,

Thank you again for submitting your manuscript "DNA-PK and the TRF2 iDDR inhibit MRN-initiated resection at leading-end telomeres". We now have comments (below) from the 3 reviewers who evaluated your paper. In light of those reports, we remain rather interested in your study and would like to see your response to the comments of the referees, in the form of a revised manuscript.

You will see that all reviewers appreciate the experimental systems used and the robustness of the data. More importantly, the referees deem that the presented conclusions are well supported and constitute an advance in our conceptual understanding. There are, however, a few points and suggestions that should be addressed in a revision. From an experimental standpoint, we would like you to address the main point of reviewer #2, with respect to solidifying and mechanistically expanding the finding that DNA-PK promotes MRN activity at telomeres, and the one of reviewer #1, with respect to strengthening the claim that alt-EJ regulates telomere fusions in the absence of Apollo.

Please be sure to address/respond to all concerns of the referees in full in a point-by-point

response and highlight all changes in the revised manuscript text file. If you have comments that are intended for editors only, please include those in a separate cover letter.

We expect to see your revised manuscript within two months. If you cannot send it within this time, please contact us to discuss an extension; we would still consider your revision, provided that no similar work has been accepted for publication at NSMB or published elsewhere.

Reporting Summary:

Please note that all key data shown in the main figures as cropped gels or blots should be presented in uncropped form, with molecular weight markers. These data can be aggregated into a single supplementary figure item. While these data can be displayed in a relatively informal style, they must refer back to the relevant figures. These data should be submitted with the final revision, as source data, prior to acceptance, but you may want to start putting it together at this point.

SOURCE DATA: we urge authors to provide, in tabular form, the data underlying the graphical representations used in figures. This is to further increase transparency in data

reporting, as detailed in this editorial (<http://www.nature.com/nsmb/journal/v22/n10/full/nsmb.3110.html>). Spreadsheets can be submitted in excel format. Only one (1) file per figure is permitted; thus, for multi-paneled figures, the source data for each panel should be clearly labeled in the Excel file; alternately the data can be provided as multiple, clearly labeled sheets in an Excel file. When submitting files, the title field should indicate which figure the source data pertains to. We encourage our authors to provide source data at the revision stage, so that they are part of the peer-review process.

Data availability: we strongly support public availability of data. All data used in accepted papers should be available via a public data repository, or alternatively, as Supplementary Information. If data can only be shared on request, please explain why in your Data Availability Statement, and also in the correspondence with your editor. Please note that for some data types, deposition in a public repository is mandatory - more information on our data deposition policies and available repositories can be found below: <https://www.nature.com/nature-research/editorial-policies/reporting-standards#availability-of-data>

[redacted]

Sincerely,

Dimitris Typas
Associate Editor
Nature Structural & Molecular Biology
ORCID: 0000-0002-8737-1319

Referee expertise:

Referee #1: DNA damage response, telomere biology, MRN complex

Referee #2: molecular mechanisms of DNA recombination, DSB end resection

Referee #3: telomere biology and DNA damage response

Reviewers' Comments:

Reviewer #1:

Remarks to the Author:

DNA-PK and the TRF2 iDDR inhibit MRN-initiated resection at leading-end telomeres - NSMB-A47273

Myler et al explore the inability to form single stranded overhangs of Apollo-deprived leading-end telomeres to show that processing of telomeric blunt ends occurs by PolQ-dependent alt-EJ, and not c-NHEJ as previously thought, and that both TRF2 iDDR domain and DNA-PK independently inhibit MRN-mediated resection at blunt-ended telomeres.

The paper is elegantly written, and the data is well controlled. By using a combination of multiple knockouts of key players in different DNA damage repair pathways, the authors show that the iDDR domain of TRF2 prevents MRN/CtIP-dependent telomere resection at the leading-strand. The authors also use Alpha Fold-Multimer and in vitro experiments to suggest that TRF2 iDDR could compete with CtIP to bind MRN, and thus prevent CtIP-dependent exonuclease activity of MRN at telomeric ends. These experiments are well controlled, with telomere resection/repair assessed by telomere fusions and single-stranded overhang formation. The manuscript is suited for publication if the authors satisfactorily consider the comments bellow.

My main comments relate to the conclusion that Nbs1 protects from leading-end telomere fusions in the absence of both Apollo and DNA-PKcs. While there is a statistically significant increase in telomere fusions with the triple knockout, only 1% of telomeres are fused, suggesting that other exonucleases are recruited to γ -telomeric blunt ends. The authors mention this possibility too lightly, both in the results and the discussion sections and it would strengthen the manuscript to have the alternatives better discussed. Also, the authors could strengthen the claim that PolQ-dependent alt-EJ mediates telomere fusions in the absence of Apollo by showing loss of single-stranded overhang.

Minor comments:

Although the authors show genetically that there is a contribution of DNA-PK to the inhibition of the resection process I am unaware that there is evidence in the literature that DNA-PK is physically associated with telomeres in the absence of Apollo. Could they somehow demonstrate this association (either by telomeric ChIP or IF)?

According to the Anand et al, Moll Cell. 2016, the endonucleolytic activity of MRN is absolutely dependent on CtIP phosphorylation and the authors use in the essay a non-phosphorylated version which should not yield a resected product. Can the authors comment on how is this possible and add in the materials and methods which version of the protein has been used.

In Figure 4C, TRF2 F120A (lack of Apollo interaction) only slightly decreases ss-telomeric overhang (~ 0.9 of TRF2 WT vs ~ 0.4 in Apollo KO). In comparison, in Figure 5b, F120A shows $\sim 70\%$ of ss-telomeric overhang reduction. Could the authors please comment the variability between these replicates and the difference compared to Apollo KO?

Please present all data uniformly by including mean and error bars in the plots showing percentage of telomere fusions, like it is presented for the overhang data. Also, please include more statistical information in respective figure legends, for example are the mean/median of 3 replicate averages? In Figure 3C for example, Apollo F/F DNA-PKs-/- +sgNBS1 condition has 9 datapoints out of 30 above 0 but the mean (not stated in the figure legend) bar equals 0.

Reviewer #2:

Remarks to the Author:

Telomeres are the natural ends of chromosomes, and hence need to keep DSB processing (resection) factors in check. The relationship of telomeres and resection factors is however complicated, as controlled resection (Apollo/Exo1) is needed to establish protective t-loops to prevent aberrant end-joining. MRN, a key factor in the response to DSBs, does not appear to resect telomeres, but how its activity is restricted was not known. Here, Myler and colleagues find that the iDDR domain of TRF2 specifically inhibits the endonuclease activity of MRN by preventing its activation by CtIP, as supported by AlphaFold modeling and biochemistry. This function resembles the role of the MIN/BAT domain of yeast Rif2, however both modules have developed independently and represent an intriguing example of parallel evolution.

In the absence of Apollo, telomere end fusion is greatly elevated. As an important side issue, in the absence of the Apollo nuclease, the authors shows that alt-EJ is responsible for aberrant end fusions as resection is inactivated. However, when the iDDR domain of TRF2 is rendered non-functional, MRN can promote resection and thus reduce chromosome fusions.

Overall, this is a very strong manuscript that combines cell biology, imaging, biochemical and computational techniques. The data are of a high quality and the key conclusions are supported by complementary approaches. The results represent a notable conceptual advance in our understanding how MRN is controlled.

A specific point:

- While the majority of the manuscript was easy to follow, one exception is the somewhat surprising effect of DNA-PK (description of Figure 3), especially considering that it was shown that DNA-PK promotes MRN activity. It is apparent that in the absence of both DNA-PK and Apollo, MRN can promote resection even when TRF is present. The authors note that the function of MRN may be structural (loading of other resection factors). The model would be testable (CtIP depletion, or MRE11 nuclease inhibitors to show that the MRN nuclease is not involved). If confirmed, this part of the story would be much less speculative.

I do not have any additional concerns about the data, and list a few suggestions below. I do not consider them essential to support the conclusions of the paper, so I will leave it up to the authors whether they wish to address them.

- The authors note (Ref12) that Exo1 extends the resection tracks of Apollo. In the context of Apollo-deficient cells, is it Exo1 that extends the resection initially started by MRN? This point could help fill a little gap in the authors' model.

- Yeast Rif2 inhibits the endonuclease activity of MRX. Somewhat paradoxically, Rif2 also strongly stimulates the ATPase of Rad50, which led to the proposal that it "discharges the ATP-bound form of Rad50", PMID: 31640985, or, in other words, makes MRX to hydrolyse ATP nonproductively. With respect to the discussion on parallel evolution it would be interesting to test whether TRF2 does the same, or whether nature found a different way.

- page 3: 3' exonuclease activity of MR  MRN was meant?

Reviewer #3:

Remarks to the Author:

This elegant study addresses several core issues of telomere biology. Using a sophisticated genetic approach whose results are further supported by biochemistry and structural prediction, the authors deciphered how TRF2 and DNA-PK synergise to protect telomeres against uncontrolled MRN-dependent resection in mammals. A key finding of this work is that TRF2 iDDR motif blocks MRN by preventing CtIP binding on Rad50, therefore shutting down MRN endonuclease activity. The convergent evolution between the iDDR motif in animals and a distinct motif in some fungi toward the same mechanism is fascinating. This is a major advance in our understanding of telomere protection and MRN function.

This work also provides important results that shed new light on previous observations:

1. Leading-to-leading telomere fusions in the absence of Apollo are mostly produced by Pol Theta mediated alternative EJ, not Lig4-dependent NHEJ. That alternative EJ can join presumably blunt ends is a striking finding and will be of interest to many.
2. DNA-PK protects telomeres against resection in Apollo-deficient cells, in line with DNA-PK known function in other systems.

In yeast, Rif2 interaction with Rad50 not only blocks Mre11 endonuclease activity but also ATM (Tel1) kinase activity (e.g. Kaizer et al. 2015 Genetics 201 p573 and Hailemariam et al. 2019 J. Biol. Chem. 294 p18846). Since iDDR and Rif2 are predicted to bind Rad50 similarly (Fig. Sup. 6d), it would be interesting to clarify to which extent TRF2 iDDR can

also impact ATM activity at telomeres. Based on exciting data (this study and Okamoto et al. 2013 Nature 494 p502), can this be ruled out?

Author Rebuttal to Initial comments

Response to referee comments

Reviewer #1:

Remarks to the Author:

DNA-PK and the TRF2 iDDR inhibit MRN-initiated resection at leading-end telomeres - NSMB-A47273

Myler et al explore the inability to form single stranded overhangs of Apollo-deprived leading-end telomeres to show that processing of telomeric blunt ends occurs by PolQ-dependent alt-EJ, and not c-NHEJ as previously thought, and that both TRF2 iDDR domain and DNA-PK independently inhibit MRN-mediated resection at blunt-ended telomeres.

The paper is elegantly written, and the data is well controlled. By using a combination of multiple knockouts of key players in different DNA damage repair pathways, the authors show that the iDDR domain of TRF2 prevents MRN/CtIP-dependent telomere resection at the leading-strand. The authors also use Alpha Fold-Multimer and in vitro experiments to suggest that TRF2 iDDR could compete with CtIP to bind MRN, and thus prevent CtIP-dependent exonuclease activity of MRN at telomeric ends. These experiments are well controlled, with telomere resection/repair assessed by telomere fusions and single-stranded overhang formation. The manuscript is suited for publication if the authors satisfactorily consider the comments below.

My main comments relate to the conclusion that Nbs1 protects from leading-end telomere fusions in the absence of both Apollo and DNA-PKcs. While there is a statistically significant increase in telomere fusions with the triple knockout, only 1% of telomeres are fused, suggesting that other exonucleases are recruited to \rightarrow -telomeric blunt ends. The authors mention this possibility too lightly, both in the results and the discussion sections and it would strengthen the manuscript to have the alternatives better discussed. Also, the authors could strengthen the claim that PolQ-dependent alt-EJ mediates telomere fusions in the absence of Apollo by showing loss of single-stranded overhang.

We thank the referee for this constructive and positive review.

We agree with the reviewer on the possibility that other nucleases are involved in the resection of telomeres in the absence of DNA-PK and we have expanded on this in the results and discussion. Indeed, we tried deletion/depletion of many other nucleases/nuclease accessory factors, including Exo1 (added to revised Extended Data Fig.2e), but we never observed an increase in leading-end fusions, probably because of redundancy. We also thank the reviewer for the suggestion of showing the loss of telomere overhang signal in Apollo-deleted cells after depletion of alt-EJ factors PolQ or Lig3 to strengthen our claim. We have now added these results in revised Fig. 2c,d. The data on Lig4 have been moved to the revised Extended Data Fig2a,b since it is (now) less relevant.

Minor comments:

Although the authors show genetically that there is a contribution of DNA-PK to the inhibition of the resection process I am unaware that there is evidence in the literature that DNA-PK is physically associated with telomeres in the absence of Apollo. Could they somehow demonstrate this association (either by telomeric ChIP or IF)?

In the discussion, we have added the citations for the first two papers showing DNA-PK association at telomeres by ChIP (Hsu et al, 1999 and D'Adda di Fagagna et al., 2001). These data were obtained in the presence of Apollo, but we think that our data presented here would suggest that this is true also in absence of Apollo. In the past, our lab has tried to assay for DNA-PK at mouse telomeres with very limited (or no) success, most likely due to the poor reagents available.

According to the Anand et al, Mol Cell. 2016, the endonucleolytic activity of MRN is absolutely dependent on CtIP phosphorylation and the authors use in the assay a non-phosphorylated version which should not yield a resected product. Can the authors comment on how is this possible and add in the materials and methods which version of the protein has been used.

We thank the reviewer for pointing this out. CtIP is purified from insect cells and is phosphorylated. We added this comment in the legend of Fig. 5 and in the Materials and Methods section.

In Figure 4C, TRF2 F120A (lack of Apollo interaction) only slightly decreases ss-telomeric overhang (~0.9 of TRF2 WT vs ~0.4 in Apollo KO). In comparison, in Figure 5b, F120A shows ~ 70% of ss-telomeric overhang reduction. Could the authors please comment the variability between these replicates and the difference compared to Apollo KO?

We have also observed such variability between different sets of experiments, and we think that this could be explained by a combination of different cell lines, time of the sample collection, and quantification/normalization. For example, Apollo deletion reduces the overhang to almost 50% in the control cell line for the Fig. 2a-b, but only to about 70% in the different cells used as control for extended Data Fig. 4b-c and to about 80% in the new Fig. 2c-d.

As for Fig 4 and 5, the samples in Fig. 4c were collected 96 hours after 4-OHT addition, while the samples in Fig. 5b were collected after 120 hours, so the phenotype could be better visible over time due to a more complete deletion of endogenous TRF2. Furthermore, in Fig. 4c the normalization using the same cells-not deleted of TRF2, whereas in Fig. 5b the normalization used TRF2-WT complemented cells.

Please present all data uniformly by including mean and error bars in the plots showing percentage of telomere fusions, like it is presented for the overhang data. Also, please include more statistical information in respective figure legends, for example are the mean/median of 3 replicate averages? In Figure 3C for example, Apollo F/F DNA-PKs-/- +sgNBS1 condition has 9 datapoints out of 30 above 0 but the mean (not stated in the figure legend) bar equals 0.

The plots for fusions/metaphases are generated by pooling the results of 3 to 4 independent experiments. Since the data do not follow a gaussian distribution and have several outliers, as "single cell analysis", we think it is better to show the median without error bars instead of the mean.

In contrast, the mean with error bars works well for the overhang data, which follow a gaussian distribution. We realized that the information was not complete, and we have now made sure that all legends clearly indicate the number of experiments, the number of samples for experiments, and the statistics used. As for Figure 3c, the median is 0 since the middle score in order of magnitude for this set of data (30) is still 0.

Reviewer #2:

Remarks to the Author:

Telomeres are the natural ends of chromosomes, and hence need to keep DSB processing (resection) factors in check. The relationship of telomeres and resection factors is however complicated, as controlled resection (Apollo/Exo1) is needed to establish protective t-loops to prevent aberrant end-joining. MRN, a key factor in the response to DSBs, does not appear to resect telomeres, but how its activity is restricted was not known. Here, Myler and colleagues find that the iDDR domain of TRF2 specifically inhibits the endonuclease activity of MRN by preventing its activation by CtIP, as supported by AlphaFold modeling and biochemistry. This function resembles the role of the MIN/BAT domain of yeast Rif2, however both modules have developed independently and represent an intriguing example of parallel evolution.

In the absence of Apollo, telomere end fusion is greatly elevated. As an important side issue, in the absence of the Apollo nuclease, the authors shows that alt-EJ is responsible for aberrant end fusions as resection is inactivated. However, when the iDDR domain of TRF2 is rendered non-functional, MRN can promote resection and thus reduce chromosome fusions.

Overall, this is a very strong manuscript that combines cell biology, imaging, biochemical and computational techniques. The data are of a high quality and the key conclusions are supported by complementary approaches. The results represent a notable conceptual advance in our understanding how MRN is controlled.

We thank referee number 2 for the valuable and positive comments on our work.

A specific point:

- While the majority of the manuscript was easy to follow, one exception is the somewhat surprising effect of DNA-PK (description of Figure 3), especially considering that it was shown that DNA-PK promotes MRN activity. It is apparent that in the absence of both DNA-PK and Apollo, MRN can promote resection even when TRF is present. The authors note that the function of MRN may be structural (loading of other resection factors). The model would be testable (CtIP depletion, or MRE11 nuclease inhibitors to show that the MRN nuclease is not involved). If confirmed, this part of the story would be much less speculative.

We agree with this comment. We tested CtIP depletion with a validated shRNA and Mre11-nuclease inhibition with Mirin in the absence of DNA-PK and/or Apollo (Figure for Referee 1). Consistent with our hypothesis that the function of MRN is more structural than enzymatic, we did not observe any leading-end telomere fusions. However, since we don't have a positive control to show that CtIP downregulation and/or Mre11 inhibition were sufficient to trigger fusions in these cells, we prefer not to show the data and leave this part speculative.

I do not have any additional concerns about the data, and list a few suggestions below. I do not consider them essential to support the conclusions of the paper, so I will leave it up the authors whether they wish to address them.

- The authors note (Ref12) that Exo1 extends the resection tracks of Apollo. In the context of Apollo-deficient cells, is it Exo1 that extends the resection initially started by MRN? This point could help fill a little gap in the authors' model.

We agree with the reviewer on this point which was similarly raised by Reviewer 1. We have tested Exo1 depletion in DNA-PK null cells after deletion of Apollo and, while we did not observe increased fusions, we saw a reduction in

the overhang, indicating that Exo1 contributes to the resection initiated by MRN. We added the results in Extended data Figure 2e-g.

- Yeast Rif2 inhibits the endonuclease activity of MRX. Somewhat paradoxically, Rif2 also strongly stimulates the ATPase of Rad50, which led to the proposal that it "discharges the ATP-bound form of Rad50", PMID: 31640985, or, in other words, makes MRX to hydrolyse ATP nonproductively. With respect to the discussion on parallel evolution it would be interesting to test whether TRF2 does the same, or whether nature found a different way.

This is a very interesting hypothesis. We tested ATP hydrolysis by purified MRN in the presence or absence of TRF2, and we indeed observe a slight increase in the ATPase activity of MRN in the presence of TRF2 (Figure for Referee 2). However, these data are still very preliminary and need more controls before being published, as for activation of the ATM kinase (below).

- page 3: 3' exonuclease activity of MR  MRN was meant?

Yes, thank you. We have fixed this mistake.

Reviewer #3:

Remarks to the Author:

This elegant study addresses several core issues of telomere biology. Using a sophisticated genetic approach whose results are further supported by biochemistry and structural prediction, the authors deciphered how TRF2 and DNA-PK synergise to protect telomeres against uncontrolled MRN-dependent resection in mammals. A key finding of this work is that TRF2 iDDR motif blocks MRN by preventing CtIP binding on Rad50, therefore shutting down MRN endonuclease activity. The convergent evolution between the iDDR motif in animals and a distinct motif in some fungi toward the same mechanism is fascinating. This is a major advance in our understanding of telomere protection and MRN function.

This work also provides important results that shed new light on previous observations:

1. Leading-to-leading telomere fusions in the absence of Apollo are mostly produced by Pol Theta mediated alternative EJ, not Lig4-dependent NHEJ. That alternative EJ can join presumably blunt ends is a striking finding and will be of interest to many.
2. DNA-PK protects telomeres against resection in Apollo-deficient cells, in line with DNA-PK known function in other systems.

In yeast, Rif2 interaction with Rad50 not only blocks Mre11 endonuclease activity but also ATM (Tel1) kinase activity (e.g. Kaizer et al. 2015 Genetics 201 p573 and Hailemariam et al. 2019 J. Biol. Chem. 294 p18846). Since iDDR and Rif2 are predicted to bind Rad50 similarly (Fig. Sup. 6d), it would be interesting to clarify to which extent TRF2 iDDR can also impact ATM activity at telomeres. Based on exciting data (this study and Okamoto et al. 2013 Nature 494 p502), can this be ruled out?

We thank Reviewer 3 for this very positive review and for raising this important question.

We fully agree that it is possible that, similar to Rif2, the iDDR could inhibit not only MRN-dependent resection, but also ATM activation. While our preliminary data both *in vivo* and *in vitro* indicate that this is the case, we are unable to answer the question definitively (see below), so we added this possibility as a discussion point that needs further investigation.

In cells, we showed the slight increase of telomeric γ -H2AX foci in cells expressing TRF2- Δ iDDR (Extended Data Fig. 3), and we now add the quantification of Chk2 phosphorylation in cells expressing TRF2- Δ iDDR, TRF2-F120A and TRF2-F120 Δ iDDR over two experiments, demonstrating that there is higher Chk2 phosphorylation in cells complemented with the TRF2-F120 Δ iDDR (new Extended Data Fig. 3c). However, the number of TIFs and Chk2 phosphorylation is not so high in cells expressing TRF2- Δ iDDR as in TRF2-deleted cells. This could be due to other mechanisms inhibiting ATM or to the fast removal of the substrate for ATM by MRN/CtIP-dependent resection. Since the lack of NBS1 compromise ATM activation *per se*, we do not have a robust genetic assay to prove one or the other hypothesis.

In parallel, we tested ATM activation by MRN on purified proteins *in vitro*. As expected, the presence of MRN triggered ATM-dependent phosphorylation of an N-terminal fragment of p53 (aa 1-102). Furthermore, p53 phosphorylation is almost abolished in the presence of TRF2 (Figure for Referee 3a), and removal of the iDDR partially reduced TRF2-dependent inhibition of ATM activation (Figure for Referee 3b). While these data would suggest that TRF2 can directly inhibit the activation of ATM by MRN through the iDDR, they are still very preliminary and at this stage we would prefer to not show them and to investigate the matter further.

[Redacted]

[Redacted]

[Redacted]

Decision Letter, first revision:

Message: Our ref: NSMB-A47273A

21st Jun 2023

Dear Dr. Lottesberger,

Thank you for submitting your revised manuscript "DNA-PK and the TRF2 iDDR inhibit MRN-initiated resection at leading-end telomeres" (NSMB-A47273A). It has now been seen by the original referees and their comments are below. The reviewers find that the paper has furthered improved in revision, and therefore we'll be happy in principle to publish it in Nature Structural & Molecular Biology, pending revisions to comply with our editorial and formatting guidelines.

Sincerely,

Dimitris Typas
Associate Editor
Nature Structural & Molecular Biology
ORCID: 0000-0002-8737-1319

Reviewer #1 (Remarks to the Author):

The authors have addressed all of the points I raised with the original submission. I am happy to recommend publication in NSMB.

Reviewer #2 (Remarks to the Author):

I thank the authors for their answers and clarifications. I am happy to recommend the manuscript for publication.

I also agree to leave the preliminary data (such as with respect to ATM activation, which will be very interesting) for a future study to be addressed properly. As it stands, the manuscript elegantly shows how telomeres keep the MRN complex in check, and what are the biological consequences when this does not happen. The study is impactful and technically very well done.

Reviewer #3 (Remarks to the Author):

The added sentence in the discussion (page 12) addresses my only concern.

Author Rebuttal, first revision:

Response to referee comments

Reviewer #1:

None

We thank again the reviewer for their positive and constructive comments and the suggestions to improve our work.

Reviewer #2:

Remarks to the Author:

I thank the authors for their answers and clarifications. I am happy to recommend the manuscript for publication.

I also agree to leave the preliminary data (such as with respect to ATM activation, which will be very interesting) for a future study to be addressed properly. As it stands, the manuscript elegantly shows how telomeres keep the MRN complex in check, and what are the biological consequences when this does not happen. The study is impactful and technically very well done.

We would like to thank the reviewer for their positive comments on our revised manuscript and these kind words.

Reviewer #3:

Remarks to the Author:

The added sentence in the discussion (page 12) addresses my only concern.

We thank again the reviewer for their very positive and kind revision of our work.

Final Decision Letter:

Message 18th Jul 2023

:

Dear Dr. Lottersberger,

We are now happy to accept your revised paper "DNA-PK and the TRF2 iDDR inhibit MRN-initiated resection at leading-end telomeres" for publication as a Article in Nature Structural & Molecular Biology.

As soon as your article is published, you can generate your shareable link by entering the DOI of your article here: <http://authors.springernature.com/share> `http://authors.springernature.com/share`. Corresponding authors will also receive an automated email with the shareable link

Your paper will be published online soon after we receive proof corrections and will appear in print in the next available issue. You can find out your date of online publication by contacting the production team shortly after sending your proof corrections. Content is published online weekly on Mondays and Thursdays, and the embargo is set at 16:00 London time (GMT)/11:00 am US Eastern time (EST) on the day of publication. Now is the time to inform your Public Relations or Press Office about your paper, as they might be

interested in promoting its publication. This will allow them time to prepare an accurate and satisfactory press release. Include your manuscript tracking number (NSMB-A47273B) and our journal name, which they will need when they contact our press office.

About one week before your paper is published online, we shall be distributing a press release to news organizations worldwide, which may very well include details of your work. We are happy for your institution or funding agency to prepare its own press release, but it must mention the embargo date and Nature Structural & Molecular Biology. If you or your Press Office have any enquiries in the meantime, please contact press@nature.com.

Please note that *Nature Structural & Molecular Biology* is a Transformative Journal (TJ). Authors may publish their research with us through the traditional subscription access route or make their paper immediately open access through payment of an article-processing charge (APC). Authors will not be required to make a final decision about access to their article until it has been accepted. <https://www.springernature.com/gp/open-research/transformative-journals> Find out more about Transformative Journals

Sincerely,

Dimitris Typas
Associate Editor
Nature Structural & Molecular Biology
ORCID: 0000-0002-8737-1319

Click here if you would like to recommend Nature Structural & Molecular Biology to your librarian:

<http://www.nature.com/subscriptions/recommend.html#forms>